# DEXR: A Unified Approach Towards Environment Agnostic Exploration

## ABSTRACT

The exploration-exploitation dilemma poses pivotal challenges in reinforcement learning (RL). While recent advances in curiosity-driven techniques have demonstrated capabilities in sparse reward scenarios, they necessitate extensive hyperparameter tuning on different types of environments and often fall short in dense reward settings. In response to these challenges, we introduce the novel **D**elayed **EX**ploration **R**einforcement Learning (DEXR) framework. DEXR adeptly curbs over-exploration and optimization instabilities issues of curiosity-driven methods, and can efficiently adapt to both dense and sparse reward environments with minimal hyperparameter tuning. This is facilitated by an auxiliary exploitation-only policy that streamlines data collection, guiding the exploration policy towards high-value regions and minimizing unnecessary exploration. Additionally, this exploration policy yields diverse, in-distribution data, and bolsters training robustness with neural network structures. We verify the efficacy of DEXR with both theoretical validations and comprehensive empirical evaluations, demonstrating its superiority in a broad range of environments.

## 1 INTRODUCTION

Reinforcement learning (RL) provides a powerful framework for training agents to perform complex tasks by maximizing cumulative rewards through trial-and-error interactions within dynamic environments. In the deep learning era, reinforcement learning has achieved state-of-the-art results across domains including game-play, robotics, and control problems. Notable examples include mastering complex games such as chess, Go, and video games (Mnih et al., 2013; 2015; Van Hasselt et al., 2016; Silver et al., 2016; 2017; Arulkumaran et al., 2019; Berner et al., 2019), acquiring skills for robots (Kober et al., 2013; Gu et al., 2017; Ahn et al., 2022), and mastering control policies for power grids (Yan & Xu, 2018). The success of RL stems from its ability to learn sophisticated behavioral policies from environment feedback, without requiring extensive human engineering or supervision.

A core challenge in RL is the exploration-exploitation dilemma - balancing exploiting existing information versus probing uncertain actions and states that may yield higher returns. Naïve random exploration methods like $\epsilon$-greedy exploit well in environments with dense reward but struggle to efficiently explore with sparse reward feedback, both in bandit and RL problem (Bubeck et al., 2012; Osband et al., 2013; Bellemare et al., 2013; Mnih et al., 2013; 2015; Van Hasselt et al., 2016). This limitation has led to research on more structured exploration techniques. Theoretical research over recent decades has led to significant progress in principled exploration algorithms. Upper confident bound (UCB) method (Auer, 2002) and its variants (Abbasi-Yadkori et al., 2011; Azar et al., 2017; Jin et al., 2018; Yang & Wang, 2020; Jin et al., 2020; Zhou et al., 2021; He et al., 2023) have achieved strong regret bounds in simplified settings by optimally balancing uncertainty-driven exploration. However, these approaches rely on assumptions on environment structures and limited function approximation, enabling tight theoretical characterization. Despite the huge contribution to the understanding of the mechanism of UCB-based exploration, the practicality of these works is limited. Practical RL problems involve complex observations requiring more expressive function approximators such as neural networks (Mnih et al., 2013; 2015; Van Hasselt et al., 2016), whose statistical properties are not yet clear. Furthermore, real-world environments have unknown dynamics and rich (often continuous) state-action spaces, unlike the simplified settings studied theoretically.

Similar to the UCB framework, empirical work has made significant progress on intrinsically motivated exploration, also referred to as intrinsic-reward-driven or curiosity-driven exploration. By providing bonus rewards for novel states or dynamics predictions, intrinsic motivation provides structured exploration guidance even in complex environments. Variants including dynamics prediction error (Oudeyer et al., 2007; Stadie et al., 2015; Pathak et al., 2017; 2019), information gains (Liu & Abbeel, 2021b;a; Kim et al., 2023), and count-based bonuses (Bellemare et al., 2016; Tang et al., 2017; Ostrovski et al., 2017; Machado et al., 2020) have proven effective across challenging domains lacking external rewards. In particular, intrinsic-reward-driven methods have achieved state-of-the-art results on exploration benchmarks like Montezuma's Revenge (Bellemare et al., 2013; 2016; Ostrovski et al., 2017; Burda et al., 2018). Nevertheless, curiosity methods usually adopt an additional hyperparameter we refer to as the *exploration factor*, for scaling the intrinsic reward to control the degree of the exploration (Bellemare et al., 2016; Ostrovski et al., 2017; Machado et al., 2020; Kim et al., 2023). Despite there are methods that do not tune such exploration factors, other factors instead have to be designed or tuned carefully as the number of parallel environments, environmental steps (for on-policy policies), discount factors, batch size, and the architecture of the network for computing intrinsic rewards (Burda et al., 2018).

However, tuning the hyperparameters of intrinsically motivated agents to balance exploration and exploitation is extremely challenging: due to the lack of theoretical understanding of neural networks, it is difficult to accurately characterize the scale of its output or its convergence rate in the face of different observations. The same intrinsic reward model could behave completely differently in different environments and different intrinsic reward models can behave differently within the same environment. One has to put extensive effort into tuning the exploration factor, to scale the intrinsic reward properly for different domains, as if the scaled intrinsic reward is too small or shrinks too quickly, the exploration will not be sufficient (Burda et al., 2018). Conversely, if it is too large or shrinking too slowly, the agent will constantly explore the novel states, getting distracted by the intrinsic rewards and failing to exploit (Taiga et al., 2021; Whitney et al., 2021; Schäfer et al., 2021; Chen et al., 2022). Furthermore, large intrinsic rewards will cause problems in the optimization process by introducing a large bias to the fitting of the neural networks and thus making the learning process hard to converge (Whitney et al., 2021). This discourages the use of large exploration factors and sacrifices the potentially better exploration. Due to all these issues, it is often the case that an exploration factor right for one domain (exhibiting sufficient but not overwhelming exploratory behavior) can induce poor performance in other environments. Enhancing the applicability and adaptability of deep curiosity-based exploration remains under-studied, but crucial for tackling complex, real-world RL problems requiring efficient learning from limited signals. In this paper, we aim to answer the following question: *Can we design an algorithm that balances exploration and exploitation properly across different types of environments with **minimal hyperparameter tuning**?*

Existing works that try to resolve this problem either focus on solving the stability of function approximation brought in by adding intrinsic reward (Schäfer et al., 2021) or attempt to deal with the constant distraction caused by the agent's curiosity (Whitney et al., 2021; Chen et al., 2022). However, they either suffer from the function approximation issue or lack an effective mechanism for cutting over-exploration.

**Our contributions**   In this paper we study RL exploration problems with a focus on enhancing accessibility, adaptability, and applicability of intrinsically motivated exploration in the face of intrinsic rewards and environments with different properties with minimal need for hyperparameter tuning. We summarize our contributions below.

- We propose a simple yet effective framework DEXR (**D**elayed **EX**ploration **R**einforcement Learning), which can be used in any intrinsically motivated off-policy RL algorithms. It enhances intrinsically motivated exploration by inducing a novel exploration paradigm, as shown in Figure 1. DEXR leverages an additional exploitation policy to guide the agent to the direction that is beneficial for completing the task and then uses the exploration policy to collect novel data to refine the exploitation policy for completing the task.

- Empirically, we exhaustively evaluate DEXR with intrinsic rewards of various types and exploration factors at different scales in a large range of environments. Through thorough benchmarking and visualization, DEXR exhibits favorable performance, due to a better balance of exploration & exploitation without being sensitive to hyperparameters.

- Theoretically, we justify the efficiency of our proposed exploration pattern by adapting DEXR to popular least square value iteration with UCB-based exploration.

This paper is organized as the following routine. We first briefly review the different types of curiosities and previous works that try to solve this problem in Section 2, and the preliminaries will be introduced in Section 3. In Section 4 and 5, we formally introduce our method, the rationale behind our design, and the theoretical guarantee, then show the experiments in comparison with others. The proof of our theoretical analysis is deferred to the appendix. We will finally provide the conclusion in Section 6.

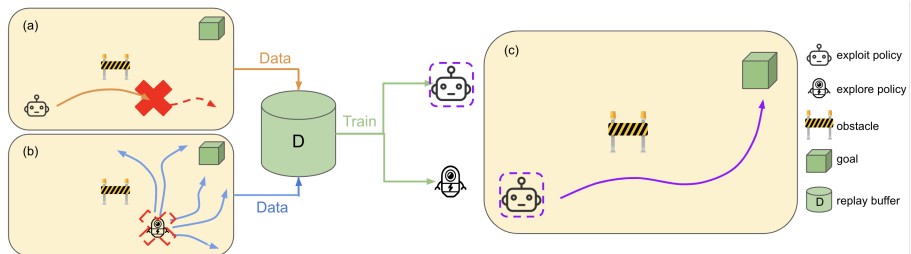

Figure 1: Illustration of DEXR. (a) Agent uses the exploitation policy stepping towards potentially promising regions based on its current knowledge. This process is truncated in the middle labelled with the solid red cross.(b) From the labelled location, the agent executes the exploration policy. (c) Data collected in (a) and (b) is used to update the policies, and exploitation policy will refine its trajectory.

## 2 RELATED WORKS

[Efficient exploration is a long-standing research topic that draws the attention of the RL community. Existing methods try to leverage the randomness in structural ways for the agent to exhibit diverse and exploratory behavior. Bayesian RL (Ghavamzadeh et al., 2015; Fortunato et al., 2017; Osband et al., 2016) leverages the ideas of Bayesian inference to quantify the uncertainty and encourage exploration accordingly. Bootstrapped DQN, the most practical variant of this line of works, wisely combines the idea of Bayesian inference and the property of the neural network for the agent to exhibit diverse and exploratory behavior, and significantly improve the performance of DQN on a handful of environments. However, it is limited to exihibit more efficient exploration on the environments that DQN already does well. Options-based methods (Sutton et al., 1999; Bacon et al., 2017; Dabney et al., 2020) are also promising for tackling the exploration problem by temporally abstracting the actions, resulting in conceptually easier policy learning and more consistent exploration, but it requires sophisticated tuning on the option switching protocol and options themselves, which limits the accessibility of this types of method.]

Intrinsic curiosity has emerged as a promising paradigm for efficient exploration in RL. There is a line of theoretical research focusing on the optimal exploration-exploitation trade-off in RL with theoretically sound intrinsic rewards. This includes (Azar et al., 2017; Yang & Wang, 2020; Jin et al., 2020; Wang et al., 2020; Ishfaq et al., 2021; Zhou et al., 2021; He et al., 2023). Yet, it is not clear how these approaches can be effectively integrated with deep networks to solve complex real-world problems.

There is also a line of works studying practical intrinsic reward, including (Oudeyer et al., 2007; Stadie et al., 2015; Pathak et al., 2017; 2019) on dynamics prediction bonuses based on model learning progress, (Bellemare et al., 2016; Ostrovski et al., 2017; Machado et al., 2020) on count-based rewards proportional to visitation density, and (Liu & Abbeel, 2021b;a; Kim et al., 2023) on entropy-based bonus, etc (Tang et al., 2017; Choshen et al., 2018; Burda et al., 2018). Dynamics-based rewards provide bonuses for improving models of environment dynamics, incentivizing visiting uncertain states (Oudeyer et al., 2007; Stadie et al., 2015; Pathak et al., 2017; 2019). Count-based methods encourage seeking rarely experienced states according to learned density models (Bellemare et al., 2016; Ostrovski et al., 2017; Machado et al., 2020). Information rewards quantify state uncertainty in an entropy-driven manner (Liu & Abbeel, 2021b;a; Kim et al., 2023). These techniques motivate exploration by quantifying different notions of novelty or uncertainty.

Recently, several works (Schäfer et al., 2021; Whitney et al., 2021; Chen et al., 2022; Li et al., 2023a) have tried to tackle the exploration problem more stably by either mitigating the hyperparameter sensitivity of intrinsically motivated exploration or designing a walk-around. (Schäfer et al., 2021) proposed to explore the environment with the exploration agent, and have the exploitation agent distill a good policy from the diverse data collected by its exploratory peer in order. The exploitation agent is only allowed to learn in a pure offline manner, which often poses heavy over-estimation problems caused by distribution shift (Fujimoto et al., 2019; Kumar et al., 2019). (Whitney et al., 2021; Chen et al., 2022) proposed to have separate exploration and exploitation policies, and let them interact with the environment alternatively. (Li et al., 2023a) proposed to solve the exploration problem without intrinsic rewards, but break down the task into easier sub-tasks for the agent to learn gradually and smoothly, however, this requires the knowledge of environment reward function and heavy effort in designing a sequence of sub-tasks. [More recently, to tackle the exploration problem in meta reinforcement learning (Meta-RL) (Norman & Clune, 2023) employs an extra exploration policy to collect diverse data for the exploitation policy to learn from. This novel method enables efficient exploration in the Meta-RL setting, but it does not reduce the (Schäfer et al., 2021; Whitney et al., 2021; Chen et al., 2022; Li et al., 2023a), but it does not solve the hyperparameter sensitivity problem.]

[(Ecoffet et al., 2019; Agarwal et al., 2020; Feng et al., 2021; Li et al., 2023b; Norman & Clune, 2023) employs extra means to relocate the agent before starting exploration. (Ecoffet et al., 2019) requires the environment to be deterministic or resettable, so the agent can accurately relate itself to states under-visited, and explore from there. Such deterministic relocation to the rarely visited states is powerful, as random exploration would suffice in this case (Ecoffet et al., 2019), however, this method requires strong environmental assumption, which is generally not accessible. (Agarwal et al., 2020; Feng et al., 2021; Li et al., 2023b) construct a set of different exploration policy sequentially for efficient exploration. Each policy is trained to explore certain areas in the environment, and before training the next policy, the agent will be relocated to the boundary between the unknown and the previously explored regions by a policy mixture of trained policies. This method avoids the over-exploration to some extent, but it still cannot overcome the distraction caused by the intrinsic reward.]

[Our proposed algorithm, DEXR, shares a similar technique of relocation with (Ecoffet et al., 2019; Agarwal et al., 2020; Feng et al., 2021; Li et al., 2023b), but with a quite different idea. Instead of relocating the agent to the uncertain regions as in (Ecoffet et al., 2019), or to the boundary as in (Agarwal et al., 2020; Feng et al., 2021; Li et al., 2023b), we employ an extra exploitation policy $\pi^{ext}$, which learns to purely exploit from the previous experience, to relocate the agent to areas that are fruitful and promising. We do so by randomly truncating the trajectory yielded by the exploitation policy, and letting the exploration policy $\pi^{int}$ explore from the truncation point. By restricting the exploration to only happen in the promising area identified by the exploitation policy, the over-exploration and distraction problem is mitigated to a large extent. And more importantly, exploratory data would be easy for the exploitation policy to digest due to its being "in-distribution" (Fujimoto et al., 2019; Kumar et al., 2019). This closed-loop enables the hyperparameter insensitive efficient online learning as we will show via the experiments and theoretical analysis.]

## 3 PRELIMINARIES

In this paper, we formulate the problem of interest as a finite horizon Markov Decision Process (MDP) (Bellman, 1957) under episodic setting, denoted by $(\mathcal{S}, \mathcal{A}, H, \gamma, r, \mathbb{P})$, where $\mathcal{S}$ is the state space, $\mathcal{A}$ is the action space, $\mathbb{P} = \{P_h\}_{h=1}^H$ are the transition operators, $r = \{r_h\}_{h=1}^H$ where $r_h : \mathcal{S} \times \mathcal{A} \to [0, 1]$ are the deterministic reward functions, $H$ is the planning horizon, i.e. episode length. $\gamma \in (0, 1]$ is the discount factor. [1]

An agent interacts with the environment episodically as follows. For each $H$-length episode, the agent adopts a policy $\pi$. To be specific, a policy $\pi : \mathcal{S} \to \mathcal{A}$ chooses an action $a$ on the action space based on the current state $s$. The policy $\pi$ induces a trajectory $s_1, a_1, r_1, s_2, a_2, r_2, \cdots s_H, a_H, r_H$, where $s_1$ is the starting point, $a_1 = \pi_1(s_1)$, $r_1 = r(s_1, a_1)$, $s_2 \sim P_1(\cdot|s_1, a_1)$, $a_2 = \pi_1(s_2)$, etc.

---

[1] In episodic setting, $\gamma$ can be set to 1 as no value explosion will happen, we will use $\gamma = 1$ in our theoretical analysis.

We use value functions, $V$-function, and $Q$-function to represent the long-term expected cumulative reward under the policy $\pi$ with respect to the current state (state-action) pair, formally defined as:

$$Q_h^\pi(s,a) = \mathbb{E}\left[\sum_{h'=h}^{H} \gamma^{h'-h} r_{h'}(s_{h'}, a_{h'}) | s_h = s, a_h = a, \pi\right] \tag{1}$$

$$V_h^\pi(s) = \mathbb{E}\left[\sum_{h'=h}^{H} \gamma^{h'-h} r_{h'}(s_{h'}, a_{h'}) | s_h = s, \pi\right], \quad \forall (s,a) \in \mathcal{S} \times \mathcal{A} \tag{2}$$

The goal of RL is to find the optimal policy $\pi^\star$ that induces the optimal value $V_h^\star(s) := \sup_\pi V_h^\pi(s)$ for any $(s,h) \in \mathcal{S} \times [H]$. We also denote $[\mathbb{P}_h V_{h+1}](s,a) := \mathbb{E}_{s' \sim \mathbb{P}_h(\cdot|s,a)} V_{h+1}(s')$, then the Bellman equation and the Bellman optimality equation can be written as :

$$Q_h^\pi(s,a) = r_h(s,a) + \gamma \mathbb{P}_h V_{h+1}^\pi(s,a) \quad \text{and} \quad V_h^\pi(s) = Q_h^\pi(s, \pi_h(s)), \quad \forall(s,a) \in \mathcal{S} \times \mathcal{A} \tag{3}$$

$$Q_h^\star(s,a) = r_h(s,a) + \gamma \mathbb{P}_h V_{h+1}^\star(s,a) \quad \text{and} \quad V_h^\star(s) = Q_h^\star(s, \pi_h^\star(s)), \quad \forall(s,a) \in \mathcal{S} \times \mathcal{A} \tag{4}$$

The underlying true value functions are unknown in RL problems, thus the agent adopts an estimated $Q$-function, which is often referred to as $Q$-estimation or estimated $Q$-value. In the function approximation setting, we denote it as $Q(s, a; \mathbf{w})$ where $\mathbf{w}$ is the parameters of this function. RL algorithms refine this $Q$-estimation function over time and use it to improve the policy in various ways. The policy can be set to deterministically pick actions maximizing the expected $Q$-estimation (Mnih et al., 2013; 2015; Van Hasselt et al., 2016), stochastically choose actions with probability proportional to $Q$-estimations of all actions (Bellemare et al., 2017; Dabney et al., 2018) when $\mathcal{A}$ is finite. Instead of setting the policy explicitly based on the $Q$-estimation in a non-parametric manner, one can also parameterize $\pi$ with a separate parameter $\theta$ to form an Actor-Critic architecture, and concurrently optimize $\theta$ to maximize the current estimated $Q$-value (Silver et al., 2014; Lillicrap et al., 2015; Mnih et al., 2016; Schulman et al., 2017; Fujimoto et al., 2018; Haarnoja et al., 2018). We formally define the policy improvement as:

$$\pi(s) \propto Q(s, \cdot\,; \mathbf{w}), \quad \pi(s) \propto_\theta Q(s, \cdot\,; \mathbf{w}) \tag{5}$$

$\propto$ denotes the greedy policy with respect to $Q$ and $\propto_\theta$ denotes update the $\theta$-parameterized policy to maximize current $Q$-estimate. In DEXR, policies are trained in an off-policy manner. In off-policy RL methods, a replay buffer $D$ is used to store the data collected by policies, and the agent uses the data to update the parameters $\mathbf{w}$ and its $Q$-function estimation. Specifically, the $\mathbf{w}$ is updated to minimize the Bellman error:

$$\mathcal{L}(\mathbf{w}) = \mathbb{E}_{(s,a,s',r) \sim D}\left[(Q(s,a;\mathbf{w}) - (r + \gamma Q(s', \pi(s'); \mathbf{w})))^2\right] \tag{6}$$

For intrinsically motivated agents with exploratory policy, equipped with an intrinsic reward model $b(s, a, s'; \kappa)$ parameterized by $\kappa$ and the exploration factor $\beta$, the augmented Bellman error is given by:

$$\mathcal{L}^{int}(\mathbf{w}) = \mathbb{E}_{(s,a,s',r) \sim D}\left[(Q(s,a;\mathbf{w}) - (r + \beta b(s,a,s';\kappa) + \gamma Q(s', \pi(s'); \mathbf{w})))^2\right] \tag{7}$$

## 4 DELAYED EXPLORATION RL

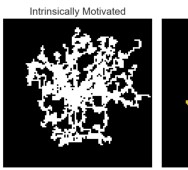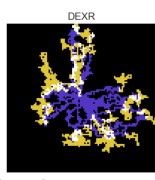

Figure 2: Exploration patterns of vanilla intrinsically motivated exploration (Left) and DEXR (Right).

DEXR algorithm utilizes two policies - an exploitation policy $\pi^{ext}$ focused solely on maximizing external task reward $r(s,a)$, and an exploration policy $\pi^{int}$ driven by the joint reward $r(s,a) + \beta b(s,a,s';\kappa)$, where $b$ is the intrinsic reward model and $\beta$ is the exploration factor for controlling the degree of exploratory behavior (Schäfer et al., 2021). To take advantage of all collected samples, both policies are trained with an off-policy RL algorithm (Mnih et al., 2013; 2015; Van Hasselt et al., 2016; Fujimoto et al., 2018; Haarnoja et al., 2018).

A generic routine of DEXR is described in Algorithm 1. The algorithm starts with two policies $\pi^{ext}, \pi^{int}$ and corresponding $Q$-functions, parameterized by $\mathbf{w}_1$ and $\mathbf{w}_2$ respectively. In each iteration, DEXR collects data and improves the policy afterward. From line 5 to line 16, the algorithm collects data and from line 17, it applies to update models and produce new policies. While the overall structure is standard, our core novelty is the structured data collection phase which we will elaborate on next.

---

**Algorithm 1** DEXR: Delayed Exploration Reinforcement Learning

---

**Require:** Horizon $H$, $Q$-functions parameterized by $\mathbf{w}_1, \mathbf{w}_2$, Policies $\pi^{ext}, \pi^{int}$, Intrinsic reward module $b$ parametrized by $\kappa$, Intrinsic reward scaling factor $\beta$, Replay buffer $D$, Truncation probability $p \in [0, 1]$

1: **for** $k = 1, 2, ..., K$ **do**
2:   Observe the initial state $s_1$
3:   Set $h \leftarrow 1$
4:   Sample truncation flag: $\text{truncate} \sim \text{Bernoulli}(p)$
5:   **while** $h < H + 1$ and $\text{truncate} \neq 1$ **do**        ▷ First phase for positioning the agent
6:     Take action $a_h = \pi^{ext}(s_h) \propto Q(s_h, \cdot\, ; \mathbf{w}_1)$ in the environment
7:     Store transition $(s_h, a_h, s_{h+1}, r_h)$ to buffer $D$
8:     Set $h \leftarrow h + 1$
9:     Update truncation flag: $\text{truncate} \sim \text{Bernoulli}(p)$
10:   **end while**
11:   **while** $h < H + 1$ **do**        ▷ Second phase for exploring promising region
12:     Take action $a_h = \pi^{int}(s_h) \propto Q(s_h, \cdot\, ; \mathbf{w}_2)$ in the environment
13:     Store transition $(s_h, a_h, s_{h+1}, r_h)$ to buffer $D$
14:     $h \leftarrow h + 1$
15:   **end while**
16:   Update $\mathbf{w}_1$ with $D$ by minimizing Equation 6
17:   Update $\mathbf{w}_2$ with $D$ by minimizing Equation 7 with intrinsic reward module $b$
18:   Improve policies $\pi^{ext}(s_h), \pi^{int}(s_h)$ with Equation 5
19:   Update the intrinsic reward: $\kappa \leftarrow \arg\min_{\kappa'} \mathbb{E}_{(s,a,s') \sim D}[b(s, a, s'; \kappa')]$
20: **end for**

---

We want to point out that the Algorithm 1 only serves as a blueprint of DEXR. One can adapt DEXR flexibly with different routines of optimization, for example, training $Q$-functions, policies, and intrinsic reward models at every environmental step, instead of training after each episode ends. One can also adapt Algorithm 1 with different off-policy RL algorithms and intrinsic rewards that might leverage different types of models and parameterizations.

In the data collection phase, DEXR provides an elegant approach to employing two distinct policies to enable more efficient exploitation & exploration. It structures each episode into two distinct, but mutually beneficial phases. In the first phase, $\pi^{ext}$ selects actions $a_t = \pi^{ext}(s_t)$ for the initial portion of the episode, allowing $\pi^{ext}$ to exploit known rewards and guide exploration towards promising areas, without getting distracted by intrinsic rewards. The first phase is truncated with probability $p$ at each step, where $p$ is a hyperparameter for controlling how long we want the exploitation policy to run. Typically, $p$ is set to be $1 - \gamma$ initially. The second phase begins at the same state and the control switches to $\pi^{int}$. For the rest of the episode, $\pi^{int}$ collects data from novel trajectories for refining the exploitation policy. [The rationale behind our design choices is stated below.]

**Design Choice: Relocate the agent using exploitation policy**   In this two-phase data collection routine, $\pi^{ext}$ and $\pi^{int}$ cooperate closely in a mutually beneficial manner. $\pi^{ext}$ serves as a means of controlling the visitation of $\pi^{int}$. Vanilla intrinsically motivated exploration explores globally, leading to very broad search trajectories expanding right from the initial distribution, this exploration pattern is prone to over-exploration, as the agent will keep expanding these exploration paths, even revisiting these paths when the intrinsic reward is too large or shrinking too slowly. Whereas in DEXR, $\pi^{int}$ performs exploration starting from where the $\pi^{ext}$ gets stopped in the first phase, resulting in tree-structured exploration paths centering the exploitation trajectories, as shown in Figure 2. In Figure 2, the agents interact with an environment without any reward, the pure intrinsically motivated exploration agent expands its exploration area board, and DEXR produces exploration trajectories (yellow) centered around the paths taken by $\pi^{ext}$ (purple), and the overlapped area is colored white. This phenomenon suggests that the $\pi^{ext}$ has a very strong control over the visitation of the $\pi^{int}$, which is the key to reducing the sensitivity to the intrinsic reward scaling hyperparameters, in the sense that DEXR focuses the exploration effort on the region that is most promising in the environment based on the agent's current experience, and thus preventing over-exploration and distraction from large intrinsic reward. Moreover, this tree-structured searching behavior will bring

data that is close to the visitation of $\pi^{ext}$, which mitigates the risk of distribution shift caused by data sharing between two policies.

**Design Choice: Truncation Probability** $p$   [The primary objective in controlling the state visitation of the exploration policy $\pi^{int}$ is to ensure that the exploitation policy $\pi^{ext}$ effectively relocates the agent to areas deemed promising. Concurrently, it's essential to avoid excessive wandering by the exploitation policy, thereby enhancing overall efficiency. A practical method to achieve this balance involves allowing the exploitation policy to operate within its 'effective horizon' $\frac{1}{1-\gamma}$, as suggested in previous study (Agarwal et al., 2020). As training progresses and the exploitation policy improves, regions proximate to the initial state become thoroughly explored. To adapt to this evolution, we gradually reduce the truncation probability $p$. This approach incrementally extends the trajectory length managed by $\pi^{ext}$, facilitating the agent's relocation to less explored states further from the initial position.]

We would like to point out that the core innovation of DEXR is the dedicated data collection routine that establishes the cooperative relationship between two policies, therefore, DEXR is able to enhance any type of intrinsically motivated exploration method.

## 5 RESULTS

### 5.1 EXPERIMENTS

In our experiments, we study how DEXR performs in terms of exploration efficiency and the exploitation capability over the vanilla intrinsically motivated exploration. We evaluate DEXR on MuJoCo simulator (Todorov et al., 2012) on various tasks with TD3 (Fujimoto et al., 2018) as the policy training method for the continuous action space accordingly. [To show the efficiency and hyperparameter insensitivity of DEXR, we evaluate DEXR with Disagreement (Pathak et al., 2019) and different $\beta$'s in this section. To further evaluate the applicability and generality of DEXR, we also evaluate DEXR with a larger set of $\beta$'s and two other different intrinsic rewards, Dynamics (Stadie et al., 2015) and ICM (Pathak et al., 2017), the results are delayed to Appendix A]. We also compare DEXR with other methods, including TD3, Exp (intrinsically motivated TD3 agent), DeRL (Schäfer et al., 2021), and EIPO (Chen et al., 2022). We will first demonstrate the results of the experiments, and then briefly discuss and analyze our insights for them. We also theoretically justify the efficiency of DEXR by adapting it to least-square value iteration and proving the convergence guarantee of DEXR. The details of implementations and analysis of the experiments are deferred to Appendix A.

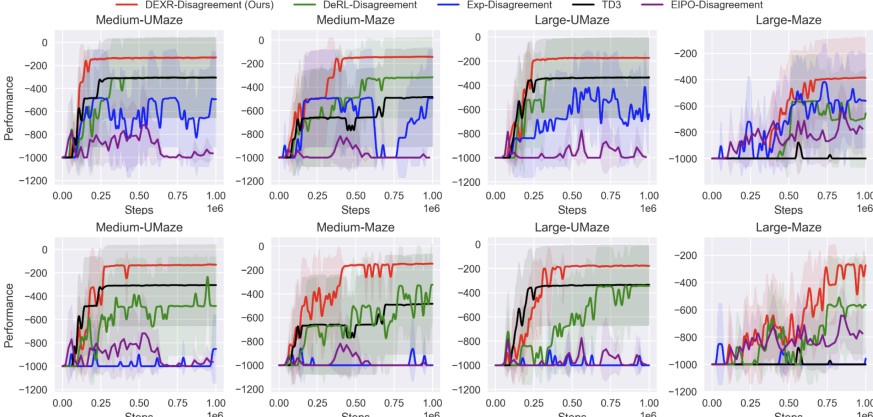

Figure 3: (Top row) Agents with small exploration factor (Bottom row) Agents with large exploration factor. Each line is averaged over 5 runs of different random seeds.

**Sparse Reward Navigation Tasks**   In the first set of experiments, we test all agents on a series of navigation tasks with different levels of difficulty, as shown in Figure 6. Within each layout, the agent starts from the green location, and the task is to transit to the red location in each episode by controlling the acceleration in both horizontal and vertical directions. In this environment, the agent

can observe its own location and velocity, but not the goal location. Each episode ends either when the horizon (set to 1000) is reached or the task is completed. The agent receives zero rewards most of the time, except a unit reward is given when it reaches the goal.

We test the agents using Disagreement intrinsic reward (Pathak et al., 2019) with two distinct hyperparameters $\beta_s = 1.0$ and $\beta_l = 10000.0$ for investigating the behavior of agents with different exploration factor [2]. The results are shown in Figure 3, where DEXR outperforms all other methods in all environments with a considerable gap with both small and overwhelmingly large exploration factors.

**DEXR balances exploration & exploitation regardless of the exploration factor** [ DEXR outperforms other agents in all navigation tasks, via a better balance of exploration & exploitation with both exploration factor $\beta_s$ and $\beta_l$, demonstrating its excellent capability of balancing exploration & exploitation and its insensitivity to exploration factor. As shown in Figure 4a, in the most difficult environment, DEXR can efficiently explore and collect data as diverse as the Exp algorithm can collect (used by DeRL), whereas TD3 fails to collect diverse data, which prevents it from consistently reaching the goal. DEXR is also capable of more efficiently leveraging the diverse dataset to learn exploitation compared to DeRL and Exp, shown in Figure 4b. ]

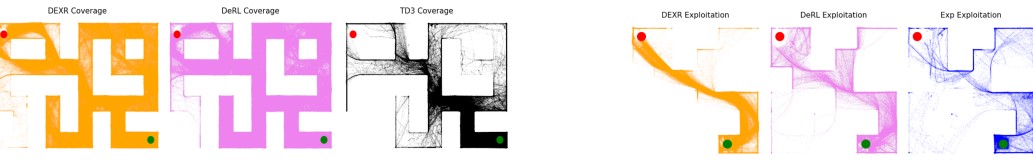

(a) Data Coverage          (b) Exploitation Trajectories

Figure 4: [(a) Visitation of DEXR compared to visitation of TD3 and DeRL in Large-Maze, DEXR collects diverse data. (b) Exploitation policy visitation of DEXR compared to DeRL and Exp, where Exp only has one policy for both exploration and exploitation.]

**Dense & Sparse Reward Locomotion** We further evaluate the efficiency and robustness of DEXR with respect to hyperparameters of the intrinsic-reward-augmented algorithms on 5 locomotion tasks in the MuJoCo simulator. In each task, the goal of the agent is to control the robot and accelerate without falling over. For a better understanding of how does DEXR perform under different reward structure, we evaluate algorithms over both dense reward and sparse reward setting, the details of the tasks are deferred to Appendix A. [In our experiment, shown in Figure 5a, DEXR performs favourably or comparable compared to Exp or TD3 (whichever performs better). And as we increase the exploration factor to $\beta_l$ to evaluate the robustness of the algorithms, DEXR shows much more robust performance compared to DeRL and Exp. Despite the performance of DEXR notably drops in some of the tasks, it performs favorably in all tasks against other intrinsic-reward driven exploration algorithms and is still able to consistently tackle the sparse Humanoid task, which is the hardest exploration task in this set of environments.]

## 5.2 THEORETICAL ANALYSIS

We show the efficiency of DEXR by showing that it enjoys a polynomial sample complexity for obtaining an $\epsilon$-optimal policy with high probability. Specifically, under the linear MDP structure, we adapt the DEXR with LSVI-UCB (Jin et al., 2020) we refer to as DEXR-UCB. With the formal structure condition described in Assumption B.1, we present our theoretical result in Theorem 5.3.

[Besides the promising performance and robustness that DEXR shows in the experiment, we would like to show it is provably efficient and can find a near-optimal policy in polynomial time. Specifically, we adapt DEXR to the least square value iteration along with UCB bonus (Jin et al., 2020), which refers to the algorithm as DEXR-UCB. DEXR-UCB is able to explore efficiently and enjoys a polynomial complexity even in the worst case with high probability. Formally, we present our result in Theorem 5.3.]

---

[2] We only use $\beta_s = 1.0$ on EIPO, as it normalizes the intrinsic reward, the detailed discussion is deferred to Appendix

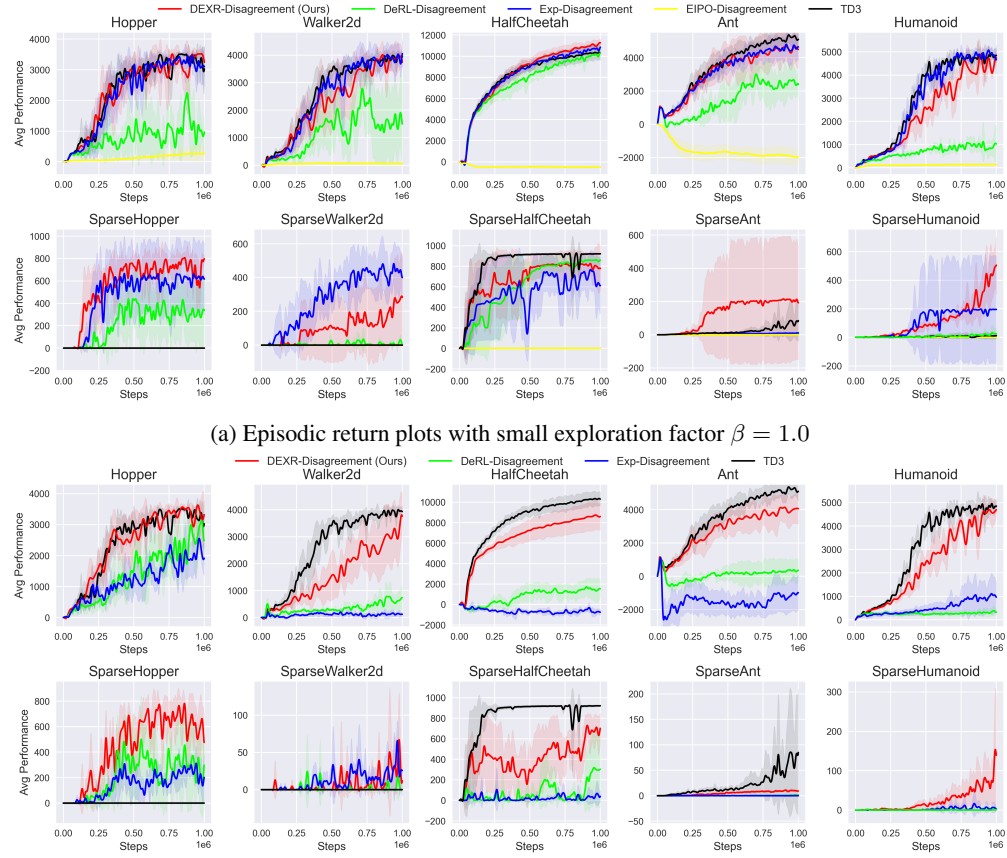

(a) Episodic return plots with small exploration factor $\beta = 1.0$

(b) Episodic return plots with small exploration factor $\beta = 1000.0$
Figure 5: MuJoCo Locomotion Performance

**Theorem 5.3.** *With horizon equal to H, DEXR-UCB learns an $\epsilon$-optimal policy within taking* $\tilde{\mathcal{O}}(\frac{d^3 H^4}{p\epsilon^2})$ *steps, with at least some constant probability.*

This result guarantees the convergence of the DEXR framework with high probability. Combined with the experiment results, Theorem 5.3 further proves the efficiency of our method. The proof of Theorem 5.3 can be found in Appendix B.

## 6 CONCLUSION & DISCUSSION

We propose DEXR, a plug-and-play framework with two distinct policies, enhancing intrinsically motivated exploration by optimizing the data collection routine. Our experiments and visualization highlight its efficiency in terms of both exploration and exploitation, as well as robustness with respect to hyperparameters. DEXR exhibits better performance, and tolerance on hyperparameters compared to existing methods consistently across various types of environments.

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

## A Implementation & Experiment Details

### A.1 Implementation of Agents

We implement DEXR, DeRL, and Exp based on the official implementation of TD3 (Fujimoto et al., 2018) along with the original hyperparameters reported in the paper. For adapting the intrinsic reward method with TD3, we update the intrinsic reward model for every environmental step using 25% of data of each batch to prevent the intrinsic reward from shrinking too fast following (Burda et al., 2018). In the experiments, we generally set the truncation probability to be $1 - \gamma$ initially. The exploitation policy will then run for $\frac{1}{1-\gamma}$ steps in expectation, where $\frac{1}{1-\gamma}$ is considered as the "effective horizon", ensuring the exploitation policy clearly points the direction for the exploration policy. Moreover, we choose to decay such $p$ over the course of training to $\frac{1}{H}$, so that the exploitation policy can more effectively control the visitation of the exploration policy as it gets improved through the training.

Continuous-action-version EIPO based on the official implementation of (Chen et al., 2022) using the open-source package (Huang et al., 2022). Further implementation details and explanations are summarized below.

- For DEXR, we initialize two TD3 policies serving for exploitation and exploration respectively. The agent first uses the exploitation policy to interact with the environment, truncated with probability $p$, then uses the exploration policy to continue the episode. Two policies share the same replay buffer and get updated at every environmental step.

- For DeRL, we initialize two TD3 policies serving for exploitation and exploration respectively. The exploration policy is used to interact with the environment and collect data, at every environmental step, both policies get updated with the collected data.

- For Exp, we use the TD3 backbone, and add the intrinsic reward to its regression target when updating $Q$-function.

- For EIPO, we adapt the official implementation to MuJoCo environments by replacing its CNN with an MLP with the same architecture used in PPO (Schulman et al., 2017) for MuJoCo experiment. And we switch its policy head from a softmax layer for discrete action space to a truncated Gaussian layer for continuous control, following the implementation of PPO.

We want to mention that, in our experiment, we only use exploration factors $\beta = 1.0$ for EIPO in all the tasks. Since EIPO uses intrinsic reward normalization by default, it does not make a difference to use $\beta$'s at different scales. In EIPO, what actually serves as the exploration factor as in other methods is the *extrinsic advantage ratio*, which is used to control the ratio of the intrinsic and extrinsic advantages for controlling the degree of the exploration behavior. This hyperparameter is set to be 2, which can be roughly considered as using exploration factor $\beta = 0.5$. The detail can be found in (Burda et al., 2018).

### A.2 PointMaze

PointMaze environment was originally designed for goal-reaching tasks (Fu et al., 2020), where an agent can observe its own location, velocity, and goal location. We slightly modify this environment by changing the observation that agent to make this environment suitable for exploration benchmarking.

For our modified PointMaze environment (Fu et al., 2020), the agent starts randomly at the green location, and its task is to reach the red location within 1000 steps for every episode. The goal location (red) is not visible to the agent, as shown in Figure 6. The agent observes a state vector that includes information on its current location and current velocity at every step, and it can control the acceleration in both horizontal and vertical directions through its action. Within this series of tasks, the agent will not receive any reward unless it reaches the goal location, in which case a unit reward will be given to the agent. This means the agent has to first explore sufficiently to find the goal location and then exploit the task efficiently by returning to the goal.

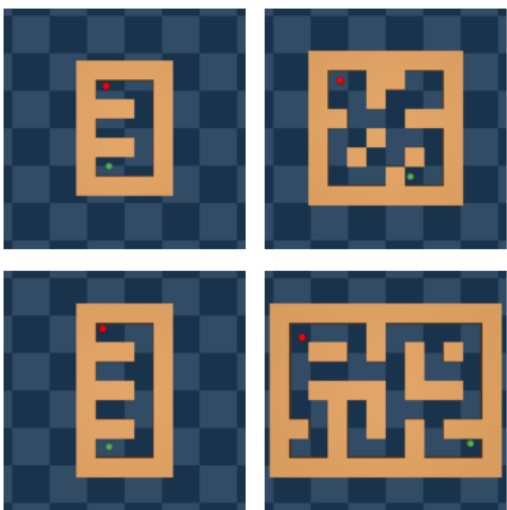

Figure 6: Layouts (Top to bottom, left to right): Medium-UMaze, Large-UMaze, Medium-Maze, Large-Maze. The green points indicate the starting location in each layout, and the red points are the goal locations.

### A.2.1 ANALYSIS ON AGENT BEHAVIORS

[DEXR outperforms other agents in all navigation tasks, via a better balance of exploration & exploitation, whereas DeRL and TD3 often come right after. TD3 relies on random noise to explore, which is inefficient. Despite tackling the task in Medium-UMaze, Large-UMaze, and Medium-Maze (suboptimally), it completely fails to solve the task in Large-Maze, the hardest one. Due to the inefficient exploration, TD3 fails to sample enough successful trajectories for it to learn and exploit, as shown in Figure 4a previously in the main text. Exp manages to explore well with a small exploration factor, solving every navigation task, but as aforementioned, optimizing the combining objective (both intrinsic and extrinsic reward) makes Exp agents hard to exploit. Exp agent gets distracted constantly during the course of learning, leading to an unstable performance with a small exploration factor, and completely fails in all the tasks with a large exploration factor. EIPO utilizes the exploration and exploitation policy alternatively, which can lead the agent to wander around, yielding poor performance in all tasks. DeRL is more robust and stable when compared to Exp and EIPO, however. However, unlike DEXR, the exploitation policy has no influence on the visitation of the exploration policy, which limits the quality of the collected data and leads to poor exploitation compared to DEXR. The visualization is shown in Figure 7. ]

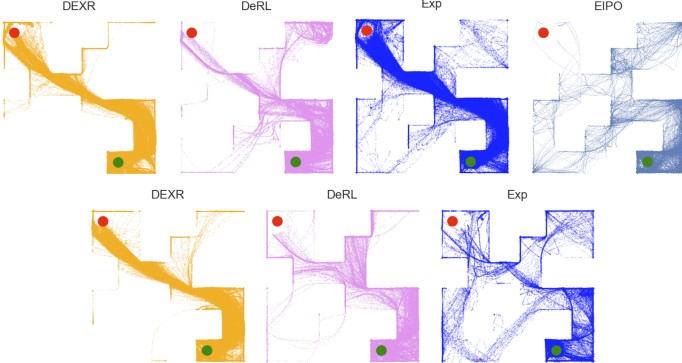

Figure 7: [We show the exploitation policy behavior of agents with the small exploration factor $\beta_s$ (Top row) and $\beta_l$ (Bottom row). The visualization is by rolling out the exploitation policy after the agent has reached the goal location once. (Orange) Exploitation policy in DEXR quickly learns to exploit regardless of the exploration factor. (Pink) DeRL is not sensitive to exploration factors but does not yield a good exploitation policy. (Blue) Exp learns well with a small exploration factor but gets heavily distracted when the exploration factor is large. (Gray) EIPO fails to accurately locate the goal using its exploitation policy.]

## A.3 MuJoCo Locomotion

For the locomotion environments (Todorov et al., 2012), the agent starts idle and the task is to control the robot to move forward as fast as possible within 1000 steps, and the episode will end if the robot falls down. The agent will observe the position, velocity, and angular velocity of the joints of the robot, and take actions to control the torque on all the joints. In the dense reward version, including Hopper, Walker2d, HalfCheetah, Ant, and Humanoid. At each step, the agent will receive a performance reward proportional to its velocity and a constant "healthy reward" if it remains in a healthy position (i.e. not falling down). In the sparse reward version, including SparseHopper, SparseWalker2d, SparseHalfCheetah, SparseAnt, and SparseHumanoid, the agent will not receive the healthy reward, and will only get a unit reward once its forward speed exceeds some threshold.

We further evaluate the efficiency and robustness with respect to hyperparameters of the intrinsic-reward-augmented algorithms by using three different intrinsic rewards (Stadie et al., 2015; Pathak et al., 2017; 2019) and five exploration factors at different scales $\beta = [1.0, 10.0, 50.0, 100.0, 1000.0]$. The aggregated results are shown in Figure 8, where each line is averaged over five agents with different exploration factors and five random seeds. For better clarity, we use red colors on DEXR's, green colors on DeRL's, blue colors on Exp, and yellow colors on EIPO, with different line styles for better distinguishing different intrinsic rewards. DEXR consistently performs favorably with different types of intrinsic rewards across various types of environments and enjoys notably smaller variances compared to DeRL and Exp in most of the environments. The split results of different $\beta's$ are shown in Figure 11, where DEXR shows more robust performance with drastically different $\beta$'s compared to DeRL and Exp. Despite the quality of the intrinsic reward still affecting the performance when using fairly coarse intrinsic reward like Dynamics (Oudeyer et al., 2007; Stadie et al., 2015), DEXR still exhibits much more consistent performance compared to DeRL and Exp.

Notably, DeRL performs even worse than Exp in most of the tasks in most of the settings, due to the distribution shift problem, as illustrated in detail in Figure 9.

Moreover, as DEXR takes in another hyperparameter, the truncation probability $p$, we aim to show that DEXR is not sensitive to this hyperparameter either. As discussed in Section 4, we typically use $1 - \gamma$ as the truncation probability and gradually decay it to $\frac{1}{H}$ as the exploitation policy gets better. We modify the truncation probability $p$, both the initial value and its decay rate, to investigate its impact as a hyperparameter. Specifically, we assess how DEXR behaves under various truncation probabilities across different environmental conditions, including dense reward and sparse reward settings. To isolate the influence of the exploration factor, we conduct our analysis using extreme values of $\beta = 1.0$ and $\beta = 1000.0$, which represent the full range we considered for MuJoCo locomotion tasks. As illustrated in Figure 10, our findings reveal that DEXR consistently performs favorably compared to Exp with the same exploration factor $\beta$, irrespective of the changes in truncation probabilities and decay configurations.

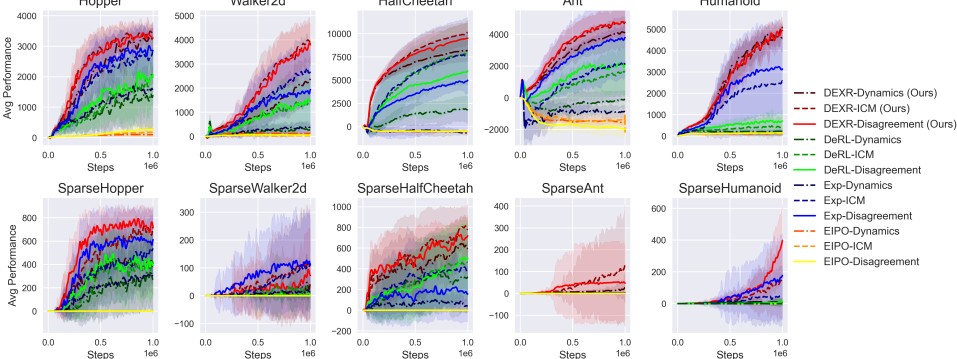

Figure 8: MuJoCo locomotion tasks with dense reward (Top row) and sparse reward (Bottom row). Performance of agents with intrinsically motivated explorations with exploration factors $\beta \in [1, 10, 50, 100, 1000]$. Each line is averaged over five different $\beta$'s with five random seeds, and DEXR shows clear improvement in terms of performance and robustness.

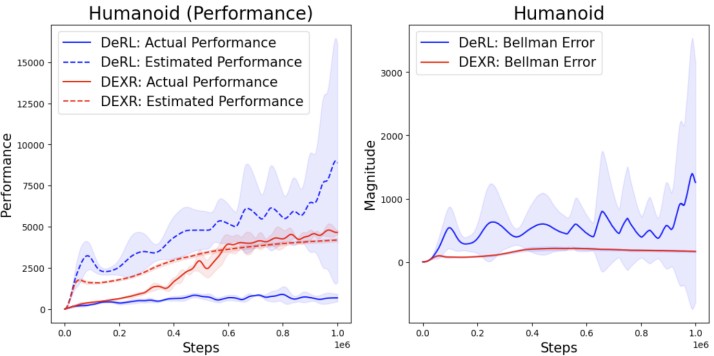

Figure 9: Distribution shift in DeRL. (Left) DeRL heavily overestimates its current performance in its $Q$-estimation. (Right) DeRL suffers from very high Bellman errors and induces unstable training. While DEXR accurately estimates the $Q$-value and enjoys smooth Bellman error.

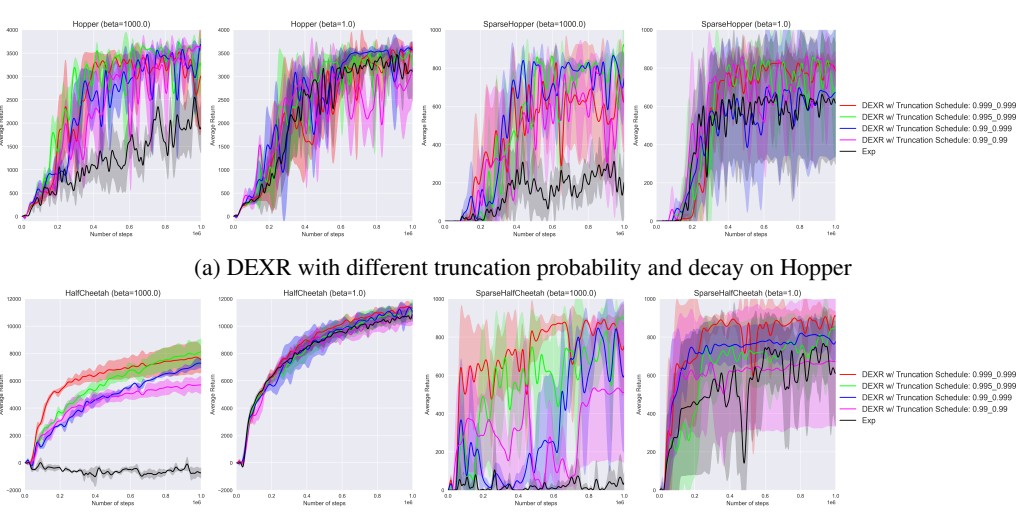

(a) DEXR with different truncation probability and decay on Hopper

(b) DEXR with different truncation probability and decay on HalfCheetah

Figure 10: DEXR is not sensitive to the truncation probability

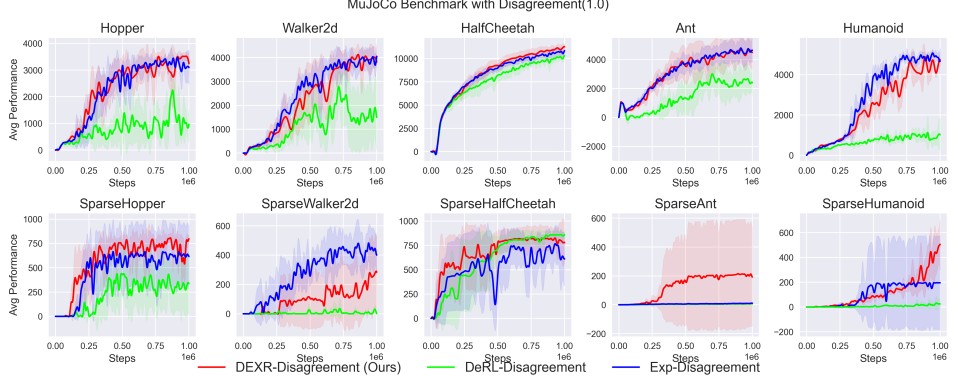

(a) Split results for DEXR, DeRL, and Exp with Disagreement ($\beta = 1.0$)

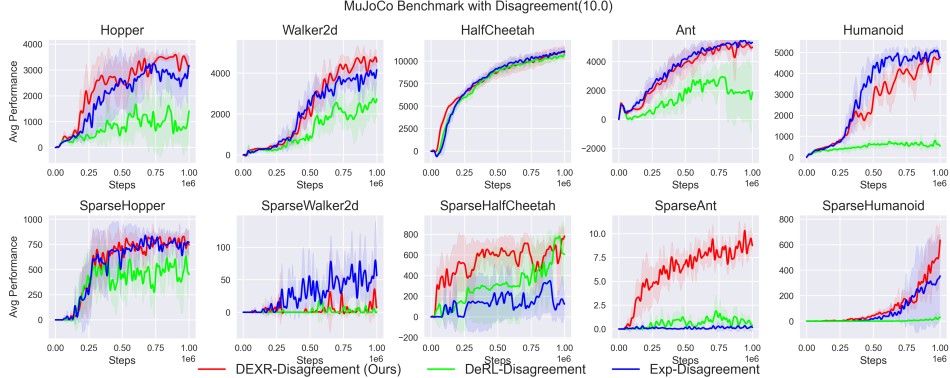

(b) Split results for DEXR, DeRL, and Exp with Disagreement ($\beta = 10.0$)

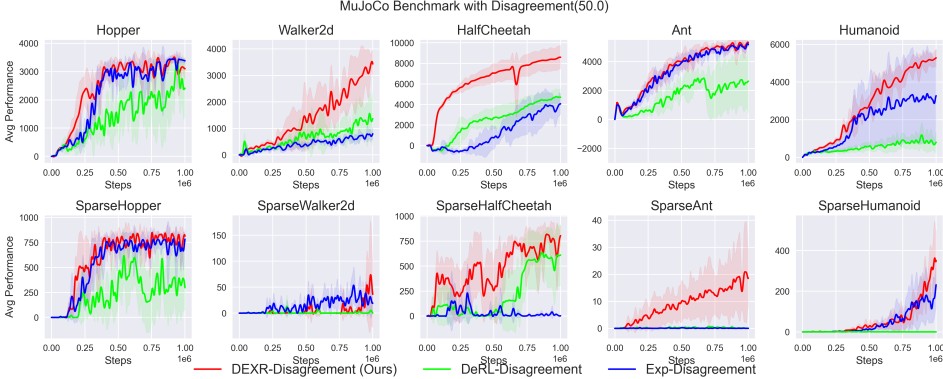

(c) Split results for DEXR, DeRL, and Exp with Disagreement ($\beta = 50.0$)

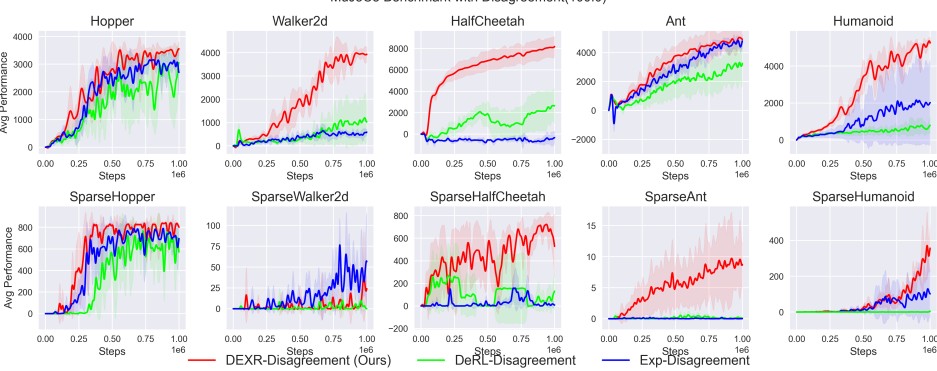

(d) Split results for DEXR, DeRL, and Exp with Disagreement ($\beta = 100.0$)

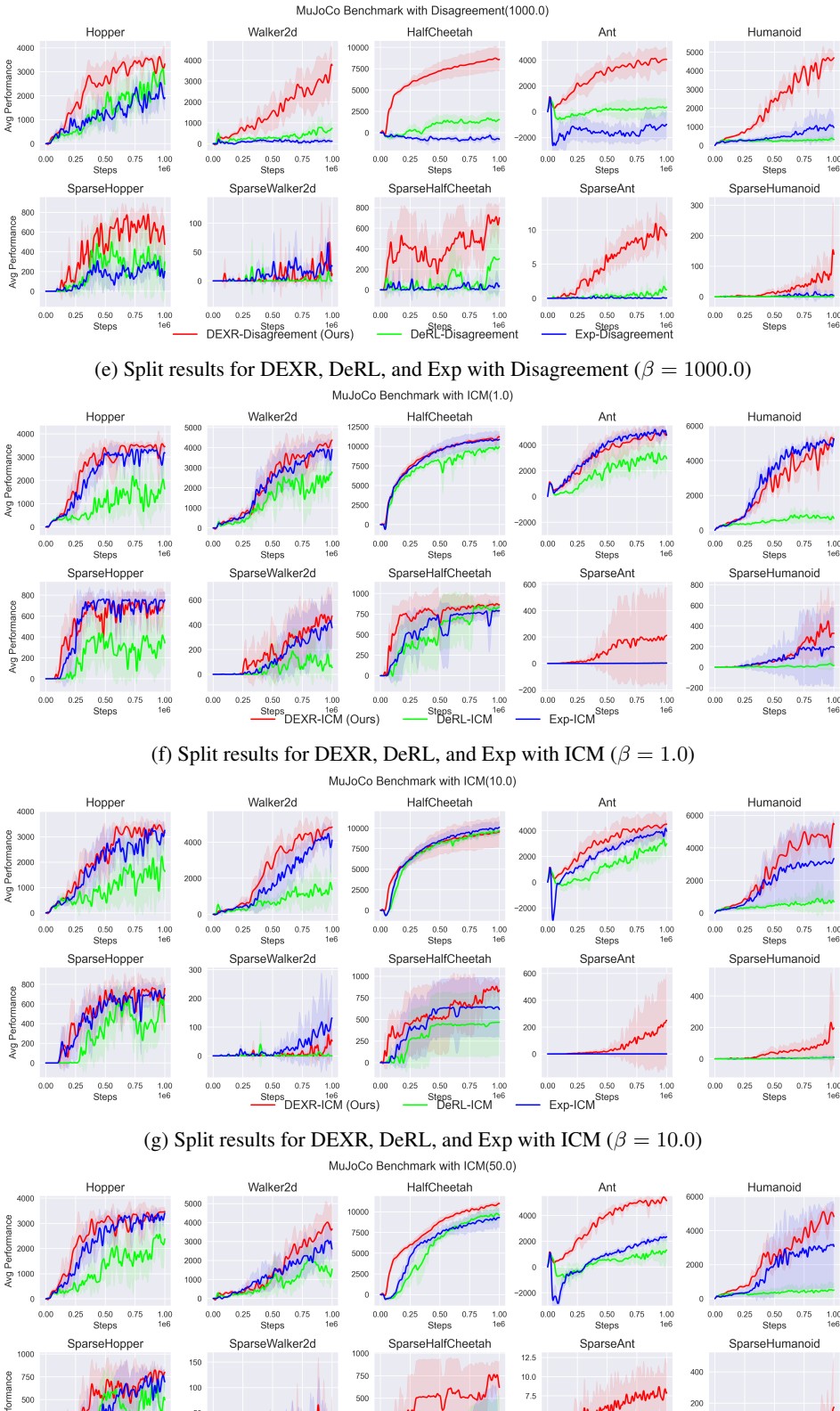

(e) Split results for DEXR, DeRL, and Exp with Disagreement ($\beta = 1000.0$)

(f) Split results for DEXR, DeRL, and Exp with ICM ($\beta = 1.0$)

(g) Split results for DEXR, DeRL, and Exp with ICM ($\beta = 10.0$)

(h) Split results for DEXR, DeRL, and Exp with ICM ($\beta = 50.0$)

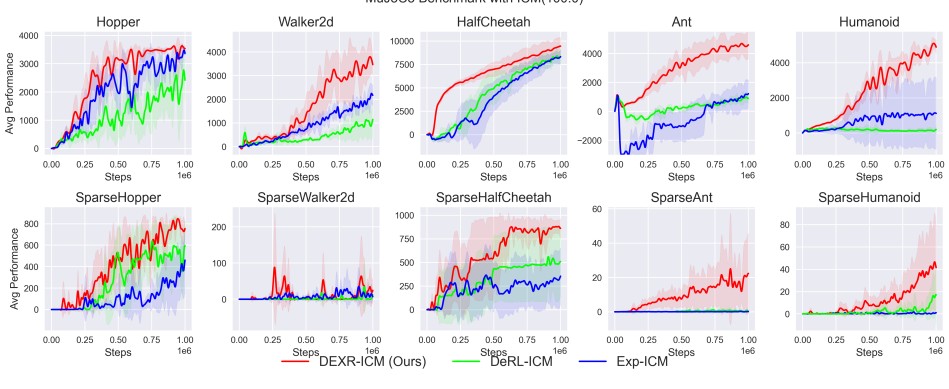

(i) Split results for DEXR, DeRL, and Exp with ICM ($\beta = 100.0$)

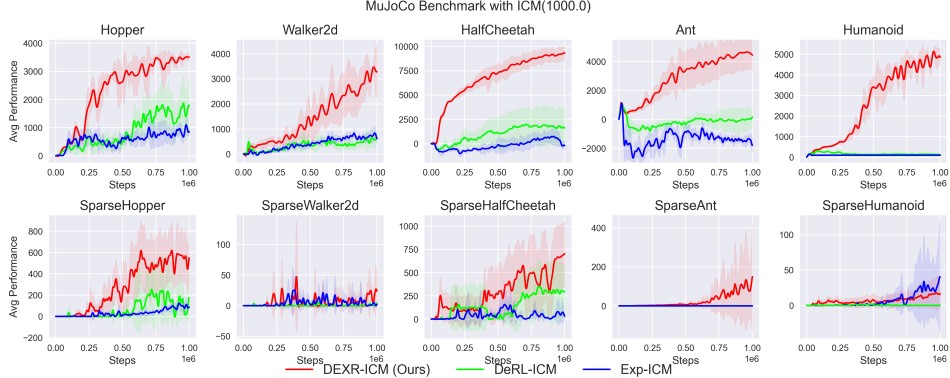

(j) Split results for DEXR, DeRL, and Exp with ICM ($\beta = 1000.0$)

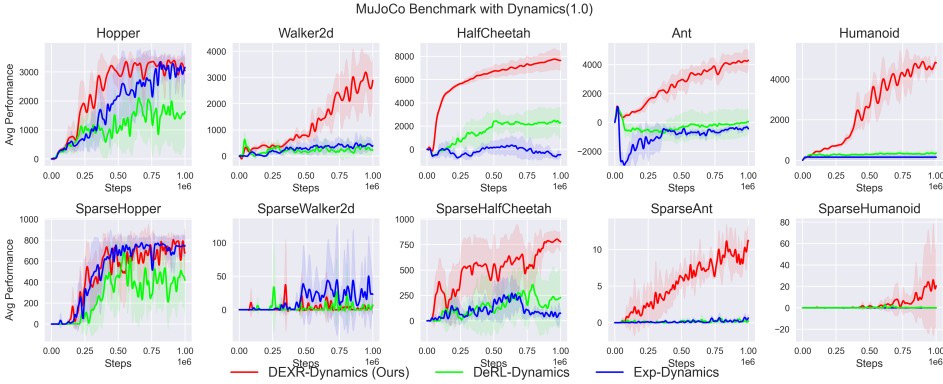

(k) Split results for DEXR, DeRL, and Exp with Dynamics ($\beta = 1.0$)

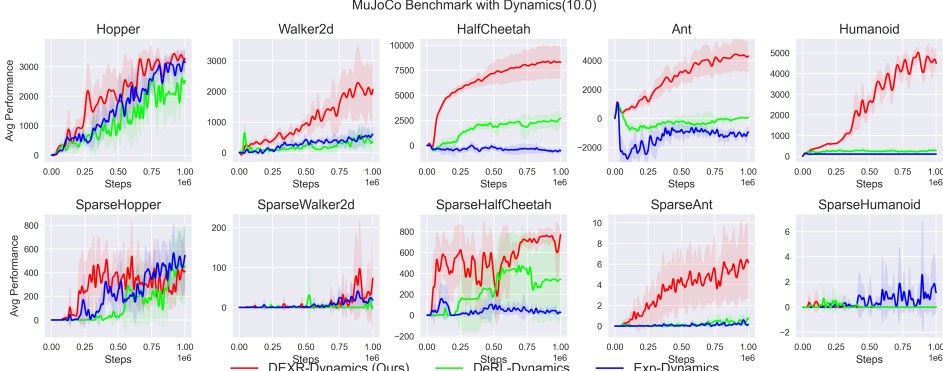

(l) Split results for DEXR, DeRL, and Exp with Dynamics ($\beta = 10.0$)

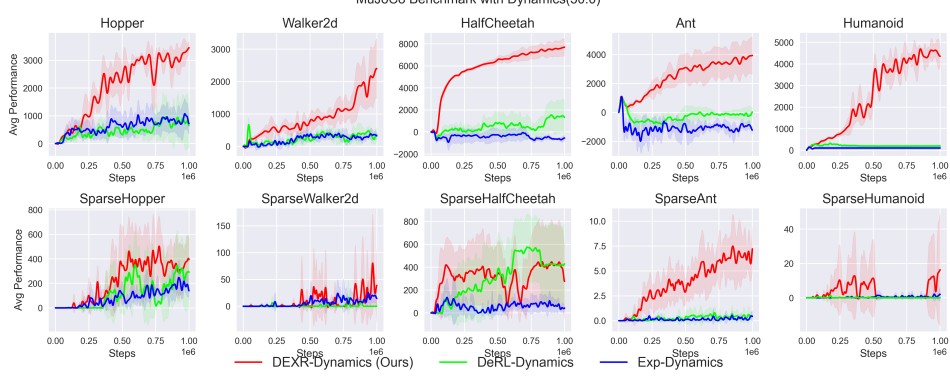

(m) Split results for DEXR, DeRL, and Exp with Dynamics ($\beta = 50.0$)

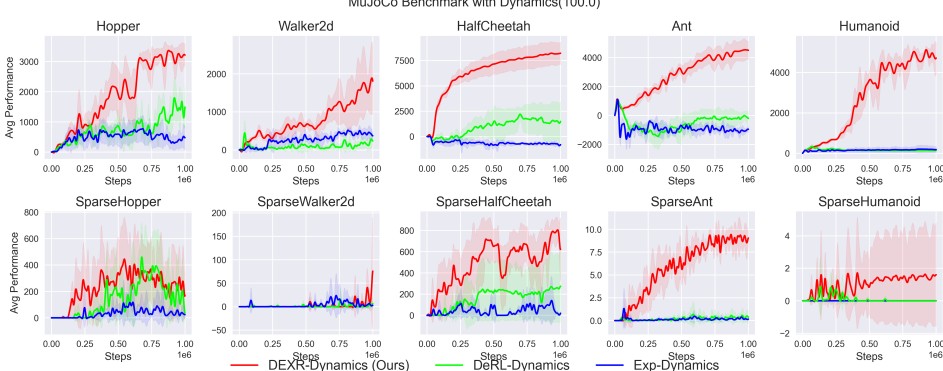

(n) Split results for DEXR, DeRL, and Exp with Dynamics ($\beta = 100.0$)

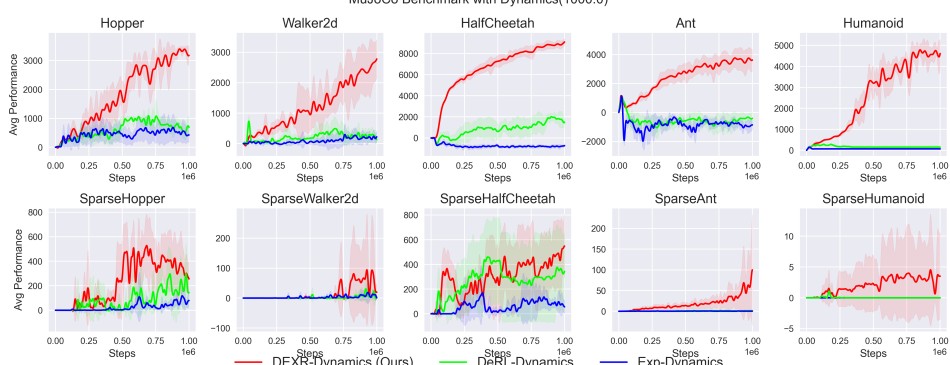

(o) Split results for DEXR, DeRL, and Exp with Dynamics ($\beta = 1000.0$)

Figure 11: MuJoCo Locomotion Split Results

## A.4 Hyperparameters of Experiments

### A.4.1 Hyperparameters for TD3-based Agents

| Hyperparameter | Value |
| --- | --- |
| Learning Rate | 3e-4 |
| Intrinsic Reward Learning Rate | 1e-4 |
| Batch Size | 256 |
| Policy Update Delay | 2 |
| Optimizer | Adam |
| $Q$-Network Architecture | (256, 256) |
| Actor-Network Architecture | (256, 256) |
| Activation function | ReLU |

### A.4.2 HYPERPARAMETERS FOR EIPO

| Hyperparameter | Value |
|---|---|
| Learning Rate | 3e-4 |
| Intrinsic Reward Learning Rate | 1e-4 |
| Batch Size | 64 |
| Number of Epochs | 10 |
| Number of Environments | 32 |
| Number of Steps | 32 |
| Optimizer | Adam |
| $V$-Network Architecture | (64, 64) |
| Actor-Network Architecture | (64, 64) |
| Activation function | ReLU |

### A.4.3 EXPLORATION FACTORS ON DIFFERENT ENVIRONMENT

| Environments | $\beta$ |
|---|---|
| PointMaze | [1.0, 10000.0] |
| Locomotion | [1.0, 10.0, 50.0, 100.0, 1000.0] |

### A.5 TRUNCATION PROBABILITY FOR DEXR

| Environments | Initial Truncation | Final Truncation |
|---|---|---|
| PointMaze (All Layouts) | 0.01 | 0.01 |
| Hopper | 0.01 | 0.001 |
| Walker2d | 0.01 | 0.001 |
| HalfCheetah | 0.01 | 0.001 |
| Ant | 0.001 | 0.001 |
| Humanoid | 0.001 | 0.001 |
| SparseHopper | 0.01 | 0.01 |
| SparseWalker2d | 0.01 | 0.01 |
| SparseHalfCheetah | 0.01 | 0.001 |
| SparseAnt | 0.01 | 0.01 |
| SparseHumanoid | 0.01 | 0.01 |

## B    PROOF OF THEOREM 5.3

In this section, we provide a comprehensive proof for Theorem 5.3. Algorithm 2 adapts the UCB-enhanced least-square value-iteration (Jin et al., 2020), a theoretically well-studied off-policy RL algorithm to our proposed DEXR framework, which we refer to as DEXR-UCB. The algorithm follows the generic DEXR framework shown in Algorithm 1, it first collects data in a two-phase manner, and updates the policies afterward. Note that DEXR-UCB adopts an additional pessimistic $Q$-function with weights $\check{\mathbf{w}}_{h=1}^{H}$, this $Q$-function is not used in DEXR-UCB, but serves as a tool for our proof. We put it in the Algorithm 2 just for the sake of clarity in terms of the definition and the update rule.

**Assumption B.1.** *(Linear MDP, e.g., (Yang & Wang, 2019; Jin et al., 2020)). MDP($\mathcal{S}, \mathcal{A}, H, \mathbb{P}, r$) is a linear MDP whose transition $\mathbb{P} := \{\mathbb{P}_h\}_{h=1}^{H}$ is not necessarily stationary. With a feature map $\phi : \mathcal{S} \times \mathcal{A} \to \mathbb{R}^d$, such that for any $h \in [H]$, there exists $d$ unknown measures $\mu_h = (\mu_h^{(1)}, \mu_h^{(2)}, \mu_h^{(3)}, ..., \mu_h^{(d)})$ over $\mathcal{S}$ and an unknown vector $\theta_h \in \mathbb{R}^d$, such that for any $(s, a) \in \mathcal{S} \times \mathcal{A}$ we have:*

$$\mathbb{P}_h(\cdot|s, a) = \phi(s, a)^T \mu_h(\cdot) \quad and \quad r_h(s, a) = \phi(s, a)^T \theta_h \tag{8}$$

*Without loss of generality, we also assume that $\|\phi(s, a)\| \leq 1$, and $\max\{\|\mu_h(\mathcal{S})\|, \|\theta_h\|\} \leq \sqrt{d}$ for all $(s, a, h) \in \mathcal{S} \times \mathcal{A} \times [H]$*

For simplicity and readability, we also denote $\mathbf{w}_h, \hat{\mathbf{w}}_h, \check{\mathbf{w}}_h$ at $k$-th episode as $\mathbf{w}_h^k, \hat{\mathbf{w}}_h^k, \check{\mathbf{w}}_h^k$, and denote $Q_h(\cdot, \cdot; \mathbf{w}_h), \hat{Q}_h(\cdot, \cdot; \mathbf{w}_h), \check{Q}_h(\cdot, \cdot; \check{\mathbf{w}}_h)$ at $k$-th episode as $Q_h^k(\cdot, \cdot), \hat{Q}_h^k(\cdot, \cdot), \check{Q}_h^k(\cdot, \cdot)$ when the context is clear.

---

**Algorithm 2** DEXR-LSVI-UCB

---

**Require:** Parameters $\lambda > 0, \mu > 0, \beta > 0, \beta' > 0$, Horizon $H$, Feature Mapping $\phi$, Truncation probability $p > 0$, Weights $\hat{\mathbf{w}}_h$ for optimistic $Q$-function, $Q$-function $\mathbf{w}_h$ for exploitation, $\check{\mathbf{w}}_h$ for pessimistic $Q$-function for all $h \in [H]$

**Require:** Clipping function $\text{clip}(x) : x \to \begin{cases} 0 & x \leq 0 \\ x & x \in (0, H) \\ H & x \geq H \end{cases}$, Geometrical distribution Geom

    **for** $k = 1, 2, ..., K$ **do**
        Receive the initial state $s_1^k$
        Sample length $L^k \sim \text{Geom}(p)$ for the first phase
        **for** step $h = 1, 2, ..., L^k$ **do**             ▷ First phase
            Take action $a_h^k \leftarrow arg\max_{a \in \mathcal{A}} Q_h(s_h^k, a; \mathbf{w}_h)$, and observe $s_{h+1}^k$
        **end for**
        **for** step $h = L^k + 1, L^k + 2, ..., H$ **do**        ▷ Second phase
            Take action $a_h^k \leftarrow arg\max_{a \in \mathcal{A}} \hat{Q}_h(s_h^k, a; \hat{\mathbf{w}}_h)$, and observe $s_{h+1}^k$
        **end for**
        **for** $h = H, H - 1, ..., 1$ **do**           ▷ Policy Improvement
            **if** $L^k = 0$ **then**
                $\Lambda_h \leftarrow \Sigma_{\tau=1}^{k-1} \phi(s_h^\tau, a_h^\tau) \phi(s_h^\tau, a_h^\tau)^T + \lambda \cdot \mathbf{I}$
                $\mathbf{w}_h \leftarrow \Lambda_h^{-1} \Sigma_{\tau=1}^{k-1} \phi(s_h^\tau, a_h^\tau)[r_h(s_h^\tau, a_h^\tau) + \max_a Q_{h+1}(s_h^{\tau+1}, a; \mathbf{w}_h)]$
                $\hat{\mathbf{w}}_h \leftarrow \Lambda_h^{-1} \Sigma_{\tau=1}^{k-1} \phi(s_h^\tau, a_h^\tau)[r_h(s_h^\tau, a_h^\tau) + \max_a \hat{Q}_{h+1}(s_h^{\tau+1}, a; \hat{\mathbf{w}}_h)]$
                $\check{\mathbf{w}}_h \leftarrow \Lambda_h^{-1} \Sigma_{\tau=1}^{k-1} \phi(s_h^\tau, a_h^\tau)[r_h(s_h^\tau, a_h^\tau) + \max_a \check{Q}_{h+1}(s_h^{\tau+1}, a; \check{\mathbf{w}}_h)]$
                $Q_h(\cdot, \cdot; \mathbf{w}_h) \leftarrow \text{clip}\left(\mathbf{w}_h^T \phi(\cdot, \cdot)\right)$
                $\hat{Q}_h(\cdot, \cdot; \hat{\mathbf{w}}_h) \leftarrow \text{clip}\left(\hat{\mathbf{w}}_h^T \phi(\cdot, \cdot) + \beta[\phi(\cdot, \cdot)^T \Lambda_h^{-1} \phi(\cdot, \cdot)]^{\frac{1}{2}}\right)$
                $\check{Q}_h(\cdot, \cdot; \check{\mathbf{w}}_h) \leftarrow \text{clip}\left(\check{\mathbf{w}}_h^T \phi(\cdot, \cdot) - \beta'[\phi(\cdot, \cdot)^T \Lambda_h^{-1} \phi(\cdot, \cdot)]^{\frac{1}{2}}\right)$
            **end if**
        **end for**
    **end for**

---

**Proposition B.2.** (($Q^\pi$ realizability (*Jin et al., 2020*)) *For a linear MDP, for any policy $\pi$, there exist weights $\{\mathbf{w}^\pi\}_{h \in [H]}$ such that for any $(s, a, h) \in \mathcal{S} \times \mathcal{A} \times [H]$, we have $Q_h^\pi(s, a) = \phi(s, a)^T \mathbf{w}_h^\pi$.*

*Proof.* By the Bellman equation we have:

$$Q_h^\pi(s, a) = r(s, a) + (\mathbb{P}_h V_{h+1}^\pi)(s, a) = \phi(s, a)^T \theta_h + \int V_{h+1}^\pi(s') \cdot \phi(s, a)^T d\mu_h(s')$$

$$= \phi(s, a)^T \cdot (\theta_h + \int V_{h+1}^\pi(s') d\mu_h(s')). \tag{9}$$

This directly shows that $Q_h^\pi$ is linear with respect to features $\phi$.      $\square$

**Lemma B.3.** *(Boundedness of $w_h^\pi$ (*Jin et al., 2020*)) Under Assumption B.1 for any fixed policy $\pi$, let $\{\mathbf{w}_h^\pi\}_{h \in [H]}$ be the weights such that $Q_h^\pi(s, a) = \langle \phi(s, a), \mathbf{w}_h^\pi \rangle$ for all $(s, a, h) \in \mathcal{S} \times \mathcal{A} \times [H]$. Then, we have*

$$\|\mathbf{w}_h^\pi\| \leq 2H\sqrt{d}, \quad \forall h \in [H]$$

*Proof. By the Bellman equation , we have:*

$$Q_h^\pi(s, a) = \left(r_h + \mathbb{P}_h V_{h+1}^\pi\right)(s, a), \quad \forall h \in [H]$$

*And by the Proposition B.2, we have:*

$$\mathbf{w}_h^\pi = \theta_h + \int V_{h+1}^\pi(s') \, d\mu_h(s')$$

*Under the normalization conditions of Assumption B.1 the reward at each step is in [0,1], we have:*

$$V_{h+1}^{\pi}(s') \leq H, \quad \forall s' \sim \mathbb{P}(\cdot|s,a)$$

*Thus, $\|\theta_h\| \leq \sqrt{d}$, and $\left\|\int V_{h+1}^{\pi}(s')\,\mathrm{d}\mu_h(s')\right\| \leq H\sqrt{d}$. This concludes the proof.*

**Lemma B.4.** *(Bound on $\hat{\mathbf{w}}_h^k$ in Algorithm 2 (Jin et al., 2020)) The weight $\hat{\mathbf{w}}_h^k$ in Algorihtm 2 satisfies:*

$$\left\|\hat{\mathbf{w}}_h^k\right\| \leq 2H\sqrt{dk/\lambda}$$

*Proof.* For simplicity, we denote the index set $\mathbf{U}^k = \{i \in [K] : L^i = 0\}$, i.e. the index of episodes in which the roll-in length is 0. For any index of episode $k \in [K]$, we denote $\lfloor k \rfloor = \max(\mathbf{U}^k)$ when $U$ is not empty, and $\lfloor k \rfloor = 0$ otherwise, i.e. the last time we encounter an episode whose roll-in length is 0. Suppose $\mathbf{v} \in \mathbb{R}^d$ is an arbitrary vector, we have:

$$\left|\mathbf{v}^\top \hat{\mathbf{w}}_h^k\right| = \left|\mathbf{v}^\top \left(\Lambda_h^{\lfloor k \rfloor}\right)^{-1} \sum_{\tau=1}^{\lfloor k \rfloor - 1} \phi_h^\tau \left[r(s_h^\tau, a_h^\tau) + \max_a \hat{Q}_{h+1}(s_{h+1}^\tau, a)\right]\right| \tag{10}$$

$$\leq \sum_{\tau=1}^{\lfloor k \rfloor - 1} \left|\mathbf{v}^\top \left(\Lambda_h^{\lfloor k \rfloor}\right)^{-1} \phi_h^\tau\right| \cdot 2H \tag{11}$$

$$\leq \sqrt{\left[\sum_{\tau=1}^{\lfloor k \rfloor - 1} \mathbf{v}^\top \left(\Lambda_h^{\lfloor k \rfloor}\right)^{-1} \mathbf{v}\right] \cdot \left[\sum_{\tau=1}^{\lfloor k \rfloor - 1} (\phi_h^\tau)^\top \left(\Lambda_h^{\lfloor k \rfloor}\right)^{-1} \phi_h^\tau\right]} \cdot 2H \tag{12}$$

$$\leq 2H\|\mathbf{v}\|\sqrt{d\lfloor k \rfloor/\lambda} \tag{13}$$

where the first step follows the algorithm construction, the second step follows directly from Cauchy–Schwarz inequality, and the last step follows from Lemma B.1, and the third step follows from the fact that $\left\|\hat{\mathbf{w}}_h^k\right\| = \max_{\mathbf{v}:\|\mathbf{v}\|=1}\left|\mathbf{v}^\top \hat{\mathbf{w}}_h^k\right|$.

This implies that $\left\|\hat{\mathbf{w}}_h^k\right\| \leq 2H\sqrt{d\lfloor k \rfloor/\lambda}$, and by definition of $\lfloor k \rfloor$, we have $\left\|\hat{\mathbf{w}}_h^k\right\| \leq 2H\sqrt{d\lfloor k \rfloor/\lambda} \leq 2H\sqrt{dk/\lambda}$, which concludes the proof.

$\square$

**Remark B.5.** *Let $\beta' = c' \cdot dH\sqrt{\log(2dT/\delta)}$ for some proper constant $c' > 0$, $\check{\mathbf{w}}_h^k \leftarrow \Lambda_h^{-1}\Sigma_{\tau=1}^{k-1}\phi(s_h^\tau, a_h^\tau)[r_h(s_h^\tau, a_h^\tau) + \max_a \check{Q}_{h+1}(s_h^{\tau+1}, a)]$, and $\check{Q}_h^k(\cdot,\cdot) \leftarrow \text{clip}\left(\{\check{\mathbf{w}}_h^T\phi(\cdot,\cdot) - \beta'[\phi(\cdot,\cdot)^T\Lambda_h^{-1}\phi(\cdot,\cdot)]^{\frac{1}{2}}\}, 0, H\right)$. By similar approach as in the proof of Lemma B.4, the weight $\check{\mathbf{w}}_h^k$ also satisfies:*

$$\left\|\check{\mathbf{w}}_h^k\right\| \leq 2H\sqrt{dk/\lambda}$$

*This result is direct, as the proof of Lemma B.4 does not leverage any property specific to $\hat{\mathbf{w}}_h^k$.*

We then define a high-probability event that bound the approximation error of our optimistic value function.

**Lemma B.6.** *(High Probability Event on Approximating Optimistic Value Function (Jin et al., 2020)) Under the setting of Theorem 5.3, let $c_\beta$ be the constant in the definition of $\beta$, such that*

$$\beta = c_\beta \cdot dH\sqrt{\log(2dT/\delta)}.$$

*There exists and an absolute constant $C$ that is independent of $c_\beta$ such that for any fixed $p \in [0, 1]$, if we let $\mathcal{E}$ be the event that:*

$$\forall (k,h) \in [K] \times [H]: \quad \left\| \sum_{\tau=1}^{k-1} \phi_h^\tau \left[ \hat{V}_{h+1}^k \left( s_{h+1}^\tau \right) - \mathbb{P}_h \hat{V}_{h+1}^k \left( s_h^\tau, a_h^\tau \right) \right] \right\|_{\left( \Lambda_h^k \right)^{-1}} \leq C \cdot dH \sqrt{\chi}$$

*where $\chi = \log \left[ 2 \left( c_\beta + 1 \right) dT/p \right]$, then $\mathbb{P}(\mathcal{E}) \geq 1 - p/2$.*

*Proof.* By Lemma B.4, we have:

$$\left\| \hat{\mathbf{w}}_h^k \right\| \leq 2H \sqrt{dk/\lambda}, \quad \forall (k,h) \in [K] \times [H]$$

Also, by the construction of $\Lambda_h^k$, its smallest eigenvalue is lower bounded by $\lambda$. Combining with Lemmas C.5 and C.7, for any fixed constant $\epsilon > 0$, we have:

$$\left\| \sum_{\tau=1}^{k-1} \phi_h^\tau \left[ \hat{V}_{h+1}^k \left( s_{h+1}^\tau \right) - \mathbb{P}_h \hat{V}_{h+1}^k \left( s_h^\tau, a_h^\tau \right) \right] \right\|_{\left( \Lambda_h^k \right)^{-1}}^2 \tag{14}$$

$$\leq 4H^2 \left[ \frac{d}{2} \log \left( \frac{k+\lambda}{\lambda} \right) + d \log \left( 1 + \frac{8H\sqrt{dk}}{\varepsilon\sqrt{\lambda}} \right) + d^2 \log \left( 1 + \frac{8d^{1/2}\beta^2}{\varepsilon^2\lambda} \right) + \log \left( \frac{2}{p} \right) \right] + \frac{8k^2\varepsilon^2}{\lambda} \tag{15}$$

By plugging in $\lambda = 1$ and $\beta = C \cdot dH \sqrt{log(2dT/\delta)}$ to this inequality, where $C$ is a positive constant independent of $c_\beta$, and picking $\epsilon = dH/k$ we have:

$$\left\| \sum_{\tau=1}^{k-1} \phi_h^\tau \left[ \hat{V}_{h+1}^k \left( s_{h+1}^\tau \right) - \mathbb{P}_h \hat{V}_{h+1}^k \left( s_h^\tau, a_h^\tau \right) \right] \right\|_{\left( \Lambda_h^k \right)^{-1}}^2 \leq C \cdot d^2 H^2 \log \left[ 2 \left( c_\beta + 1 \right) dT/p \right],$$

This concludes the proof. $\qquad\qquad\qquad\qquad\qquad\qquad\qquad\qquad\qquad\qquad\qquad\qquad\qquad\qquad\qquad\square$

Lemma B.6 provides the bound on the approximation of the optimistic value function, we can then bound the pessimistic value function in a similar way, as by Lemma C.8, these two functions classes share the same upper bound on the covering number.

**Lemma B.7.** *(High Probability Event on Approximating Pessimistic Value Function) Under the setting of Theorem 5.3, let $c_{\beta'}$ be the constant in the definition of $\beta'$, such that*

$$\beta' = c_{\beta'} \cdot dH \sqrt{log(2dT/\delta)}.$$

*There exists and an absolute constant $C'$ that is independent of $c_{\beta'}$ such that for any fixed $p \in [0,1]$, if we let $\mathcal{E}$ be the event that:*

$$\forall (k,h) \in [K] \times [H]: \quad \left\| \sum_{\tau=1}^{k-1} \phi_h^\tau \left[ \check{V}_{h+1}^k \left( s_{h+1}^\tau \right) - \mathbb{P}_h \check{V}_{h+1}^k \left( s_h^\tau, a_h^\tau \right) \right] \right\|_{\left( \Lambda_h^k \right)^{-1}} \leq C' \cdot dH \sqrt{\chi}$$

*where $\chi = \log \left[ 2 \left( c_{\beta'} + 1 \right) dT/p \right]$, then $\mathbb{P}(\mathcal{E}) \geq 1 - p/2$.*

*Proof.* By Lemma B.4, we have:

$$\left\| \check{\mathbf{w}}_h^k \right\| \leq 2H \sqrt{dk/\lambda}, \quad \forall (k,h) \in [K] \times [H]$$

Also, by the construction of $\Lambda_h^k$, its smallest eigenvalue is lower bounded by $\lambda$. Combining with Lemmas C.5 and C.8, for any fixed constant $\epsilon > 0$, we have:

$$\left\| \sum_{\tau=1}^{k-1} \phi_h^\tau \left[ \check{V}_{h+1}^k \left( s_{h+1}^\tau \right) - \mathbb{P}_h \check{V}_{h+1}^k \left( s_h^\tau, a_h^\tau \right) \right] \right\|_{\left( \Lambda_h^k \right)^{-1}}^2 \tag{16}$$

$$\leq 4H^2 \left[ \frac{d}{2} \log \left( \frac{k+\lambda}{\lambda} \right) + d \log \left( 1 + \frac{8H\sqrt{dk}}{\varepsilon\sqrt{\lambda}} \right) + d^2 \log \left( 1 + \frac{8d^{1/2}\beta^2}{\varepsilon^2\lambda} \right) + \log \left( \frac{2}{p} \right) \right] + \frac{8k^2\varepsilon^2}{\lambda} \tag{17}$$

By plugging in $\lambda = 1$ and $\beta' = C' \cdot dH\sqrt{log(2dT/\delta)}$ to this inequality, where $C'$ is a positive constant independent of $c_{\beta'}$, and picking $\epsilon = dH/k$ we have:

$$\left\| \sum_{\tau=1}^{k-1} \phi_h^{\tau} \left[ \check{V}_{h+1}^k \left( s_{h+1}^{\tau} \right) - \mathbb{P}_h \check{V}_{h+1}^k \left( s_h^{\tau}, a_h^{\tau} \right) \right] \right\|_{\left(\Lambda_h^k\right)^{-1}}^2 \leq C' \cdot d^2 H^2 \log \left[ 2 \left( c_\beta + 1 \right) dT/p \right],$$

This concludes the proof. $\qquad\square$

**Lemma B.8.** *(Optimistic Policy Action-Value Estimation Error ([Jin et al., 2020](#))) There exists an absolute constant $c_\beta$ such that for $\beta = c_\beta \cdot dH\sqrt{\log(2dT/p)}$, and for any fixed policy $\pi$, on the high-probability event $\mathcal{E}$ defined in Lemma [B.6](#) we have for all $(s, a, h, k) \in \mathcal{S} \times \mathcal{A} \times [H] \times [K]$ that:*

$$\left\langle \phi(s, a), \hat{\mathbf{w}}_h^k \right\rangle - Q_h^\pi(s, a) = \mathbb{P}_h \left( \hat{V}_{h+1}^k - V_{h+1}^\pi \right)(s, a) + \Delta_h^k(s, a),$$

*for some $\Delta_h^k(s, a)$ that satisfies $\left| \Delta_h^k(s, a) \right| \leq \beta \sqrt{\phi(s, a)^\top \left(\Lambda_h^k\right)^{-1} \phi(s, a)}.$*

*Proof.* By Proposition [B.2](#) and the Equation [3](#), we know for any $(s, a, h) \in \mathcal{S} \times \mathcal{A} \times [H]$ :
$$Q_h^\pi(s, a) := \left\langle \phi(s, a), \mathbf{w}_h^\pi \right\rangle = \left( r_h + \mathbb{P}_h V_{h+1}^\pi \right)(s, a)$$

And the residual between $\hat{\mathbf{w}}_h^k, \mathbf{w}_h^\pi$ is given by and can be decomposed as the following:

$$\hat{\mathbf{w}}_h^k - \mathbf{w}_h^\pi = \left(\Lambda_h^k\right)^{-1} \sum_{\tau=1}^{k-1} \phi_h^\tau \left[ r_h^\tau + \hat{V}_{h+1}^k \left( s_{h+1}^\tau \right) \right] - \mathbf{w}_h^\pi \tag{18}$$

$$= \left(\Lambda_h^k\right)^{-1} \left\{ -\lambda \mathbf{w}_h^\pi + \sum_{\tau=1}^{k-1} \phi_h^\tau \left[ \hat{V}_{h+1}^k \left( s_{h+1}^\tau \right) - \mathbb{P}_h V_{h+1}^\pi \left( s_h^\tau, a_h^\tau \right) \right] \right\} \tag{19}$$

$$= \underbrace{-\lambda \left(\Lambda_h^k\right)^{-1} \mathbf{w}_h^\pi}_{\mathbf{q}_1} + \underbrace{\left(\Lambda_h^k\right)^{-1} \sum_{\tau=1}^{k-1} \phi_h^\tau \left[ \hat{V}_{h+1}^k \left( s_{h+1}^\tau \right) - \mathbb{P}_h \hat{V}_{h+1}^k \left( s_h^\tau, a_h^\tau \right) \right]}_{\mathbf{q}_2} \tag{20}$$

$$+ \underbrace{\left(\Lambda_h^k\right)^{-1} \sum_{\tau=1}^{k-1} \phi_h^\tau \mathbb{P}_h \left( \hat{V}_{h+1}^k - V_{h+1}^\pi \right) \left( s_h^\tau, a_h^\tau \right)}_{\mathbf{q}_3}. \tag{21}$$

Now, we bound the terms on the right-hand side individually. For the first term,

$$\left| \left\langle \phi(s, a), \mathbf{q}_1 \right\rangle \right| = \left| \lambda \left\langle \phi(s, a), \left(\Lambda_h^k\right)^{-1} \mathbf{w}_h^\pi \right\rangle \right| \leq \sqrt{\lambda} \left\| \mathbf{w}_h^\pi \right\| \sqrt{\phi(s, a)^\top \left(\Lambda_h^k\right)^{-1} \phi(s, a)}.$$

For the second term, given the event $\mathcal{E}$ defined in Lemma [B.6](#), we have:

$$\left| \left\langle \phi(s, a), \mathbf{q}_2 \right\rangle \right| \leq c_0 \cdot dH\sqrt{\chi} \sqrt{\phi(s, a)^\top \left(\Lambda_h^k\right)^{-1} \phi(s, a)}$$

for an absolute constant $c_0$ independent of $c_\beta$, and $\chi = \log \left[ 2 \left( c_\beta + 1 \right) dT/p \right]$. For the third term,

$$\left\langle \phi(s, a), \mathbf{q}_3 \right\rangle = \left\langle \phi(s, a), \left(\Lambda_h^k\right)^{-1} \sum_{\tau=1}^{k-1} \phi_h^\tau \mathbb{P}_h \left( \hat{V}_{h+1}^k - V_{h+1}^\pi \right) \left( x_h^\tau, a_h^\tau \right) \right\rangle \tag{22}$$

$$= \left\langle \phi(s, a), \left(\Lambda_h^k\right)^{-1} \sum_{\tau=1}^{k-1} \phi_h^\tau \left(\phi_h^\tau\right)^\top \int \left( \hat{V}_{h+1}^k - V_{h+1}^\pi \right) \left( x' \right) \mathrm{d}\boldsymbol{\mu}_h \left( x' \right) \right\rangle \tag{23}$$

$$= \underbrace{\left\langle \phi(s, a), \int \left( \hat{V}_{h+1}^k - V_{h+1}^\pi \right) \left( x' \right) \mathrm{d}\boldsymbol{\mu}_h \left( x' \right) \right\rangle}_{p_1} \underbrace{- \lambda \left\langle \phi(s, a), \left(\Lambda_h^k\right)^{-1} \int \left( \hat{V}_{h+1}^k - V_{h+1}^\pi \right) \left( x' \right) \mathrm{d}\boldsymbol{\mu}_h \left( x' \right) \right\rangle}_{p_2}, \tag{24}$$

where, by Assumption B.1 Equation 8, we have

$$p_1 = \mathbb{P}_h \left( \hat{V}_{h+1}^k - V_{h+1}^\pi \right)(s,a), \quad |p_2| \le 2H\sqrt{d\lambda}\sqrt{\phi(s,a)^\top \left(\Lambda_h^k\right)^{-1}\phi(s,a)}$$

Finally, since $\left\langle \phi(s,a), \hat{\mathbf{w}}_h^k \right\rangle - Q_h^\pi(s,a) = \left\langle \phi(s,a), \hat{\mathbf{w}}_h^k - \mathbf{w}_h^\pi \right\rangle = \left\langle \phi(s,a), \mathbf{q}_1 + \mathbf{q}_2 + \mathbf{q}_3 \right\rangle$, by Lemma B.3 and our choice of parameter $\lambda$, we have

$$\left| \left\langle \phi(s,a), \hat{\mathbf{w}}_h^k \right\rangle - Q_h^\pi(s,a) - \mathbb{P}_h \left( \hat{V}_{h+1}^k - V_{h+1}^\pi \right)(s,a) \right| \le c' \cdot dH\sqrt{\chi}\sqrt{\phi(s,a)^\top \left(\Lambda_h^k\right)^{-1}\phi(s,a)},$$

for an absolute constant $c'$ independent of $c_\beta$. Finally, to prove this lemma, we only need to show that there exists a choice of absolute constant $c_\beta$ so that

$$c'\sqrt{\iota + \log(c_\beta + 1)} \le c_\beta \sqrt{\iota} \tag{25}$$

where $\iota = \log(2dT/p)$. We know $\iota \in [\log 2, \infty)$ by its definition, and $c'$ is an absolute constant independent of $c_\beta$. Therefore, we can pick an absolute constant $c_\beta$ which satisfies $c'\sqrt{\log 2 + \log(c_\beta + 1)} \le c_\beta \sqrt{\log 2}$. This choice of $c_\beta$ will make Equation 25 hold for all $\iota \in [\log 2, \infty)$, which finishes the proof.

$\square$

With similar approach, we can bound the action-value approximation for the pessimistic policy.

**Lemma B.9.** *(Pessimistic Policy Action-Value Estimation Error) There exists an absolute constant $c_{\beta'}$ such that for $\beta' = c_{\beta'} \cdot dH\sqrt{\log(2dT/p)}$, and for any fixed policy $\pi$, on the high-probability event $\mathcal{E}$ defined in Lemma B.7 we have for all $(s,a,h,k) \in \mathcal{S} \times \mathcal{A} \times [H] \times [K]$ that:*

$$\left\langle \phi(s,a), \check{\mathbf{w}}_h^k \right\rangle - Q_h^\pi(s,a) = \mathbb{P}_h \left( \check{V}_{h+1}^k - V_{h+1}^\pi \right)(s,a) + \tilde{\Delta}_h^k(s,a),$$

*for some $\tilde{\Delta}_h^k(s,a)$ that satisfies $\left| \tilde{\Delta}_h^k(s,a) \right| \le \beta' \sqrt{\phi(s,a)^\top \left(\Lambda_h^k\right)^{-1}\phi(s,a)}$.*

*Proof.* Similar to the proof of Lemma B.8, we decompose the residule between $\check{\mathbf{w}}_h^k$ and $\mathbf{w}_h^\pi$ as the following:

$$\check{\mathbf{w}}_h^k - \mathbf{w}_h^\pi = \left(\Lambda_h^k\right)^{-1} \sum_{\tau=1}^{k-1} \phi_h^\tau \left[ r_h^\tau + \check{V}_{h+1}^k \left(s_{h+1}^\tau\right) \right] - \mathbf{w}_h^\pi \tag{26}$$

$$= \left(\Lambda_h^k\right)^{-1} \left\{ -\lambda \mathbf{w}_h^\pi + \sum_{\tau=1}^{k-1} \phi_h^\tau \left[ \check{V}_{h+1}^k \left(s_{h+1}^\tau\right) - \mathbb{P}_h V_{h+1}^\pi \left(s_h^\tau, a_h^\tau\right) \right] \right\} \tag{27}$$

$$= \underbrace{-\lambda \left(\Lambda_h^k\right)^{-1} \mathbf{w}_h^\pi}_{\mathbf{q}_1} + \underbrace{\left(\Lambda_h^k\right)^{-1} \sum_{\tau=1}^{k-1} \phi_h^\tau \left[ \check{V}_{h+1}^k \left(s_{h+1}^\tau\right) - \mathbb{P}_h \check{V}_{h+1}^k \left(s_h^\tau, a_h^\tau\right) \right]}_{\mathbf{q}_2} \tag{28}$$

$$+ \underbrace{\left(\Lambda_h^k\right)^{-1} \sum_{\tau=1}^{k-1} \phi_h^\tau \mathbb{P}_h \left( \check{V}_{h+1}^k - V_{h+1}^\pi \right)\left(s_h^\tau, a_h^\tau\right)}_{\mathbf{q}_3}. \tag{29}$$

By the proof of the first term,

$$|\langle \phi(s,a), \mathbf{q}_1 \rangle| = \left| \lambda \left\langle \phi(s,a), \left(\Lambda_h^k\right)^{-1} \mathbf{w}_h^\pi \right\rangle \right| \le \sqrt{\lambda} \|\mathbf{w}_h^\pi\| \sqrt{\phi(s,a)^\top \left(\Lambda_h^k\right)^{-1}\phi(s,a)}.$$

For the second term, given the event $\mathcal{E}$ defined in Lemma B.7, we have:

$$|\langle \phi(s,a), \mathbf{q}_2 \rangle| \leq c_0 \cdot dH\sqrt{\chi}\sqrt{\phi(s,a)^\top \left(\Lambda_h^k\right)^{-1}\phi(s,a)}$$

for an absolute constant $c_0$ independent of $c_\beta$, and $\chi = \log\left[2\left(c_{\beta'}+1\right)dT/p\right]$. For the third term,

$$\langle \phi(s,a), \mathbf{q}_3 \rangle = \left\langle \phi(s,a), \left(\Lambda_h^k\right)^{-1}\sum_{\tau=1}^{k-1}\phi_h^\tau \mathbb{P}_h\left(\check{V}_{h+1}^k - V_{h+1}^\pi\right)(x_h^\tau, a_h^\tau)\right\rangle \tag{30}$$

$$= \left\langle \phi(s,a), \left(\Lambda_h^k\right)^{-1}\sum_{\tau=1}^{k-1}\phi_h^\tau \left(\phi_h^\tau\right)^\top \int \left(\check{V}_{h+1}^k - V_{h+1}^\pi\right)(x')\,\mathrm{d}\boldsymbol{\mu}_h(x')\right\rangle \tag{31}$$

$$= \underbrace{\left\langle \phi(s,a), \int \left(\check{V}_{h+1}^k - V_{h+1}^\pi\right)(x')\,\mathrm{d}\boldsymbol{\mu}_h(x')\right\rangle}_{p_1} \tag{32}$$

$$\underbrace{-\lambda\left\langle \phi(s,a), \left(\Lambda_h^k\right)^{-1}\int \left(\check{V}_{h+1}^k - V_{h+1}^\pi\right)(x')\,\mathrm{d}\boldsymbol{\mu}_h(x')\right\rangle}_{p_2} \tag{33}$$

where, by Equation (3), we have

$$p_1 = \mathbb{P}_h\left(\check{V}_{h+1}^k - V_{h+1}^\pi\right)(s,a), \quad |p_2| \leq 2H\sqrt{d\lambda}\sqrt{\phi(s,a)^\top \left(\Lambda_h^k\right)^{-1}\phi(s,a)}$$

Finally, since $\left\langle \phi(s,a), \check{\mathbf{w}}_h^k\right\rangle - Q_h^\pi(s,a) = \left\langle \phi(s,a), \check{\mathbf{w}}_h^k - \mathbf{w}_h^\pi\right\rangle = \langle \phi(s,a), \mathbf{q}_1 + \mathbf{q}_2 + \mathbf{q}_3\rangle$, by Lemma B.3 and our choice of parameter $\lambda$, we have

$$\left|\left\langle \phi(s,a), \check{\mathbf{w}}_h^k\right\rangle - Q_h^\pi(s,a) - \mathbb{P}_h\left(\check{V}_{h+1}^k - V_{h+1}^\pi\right)(s,a)\right| \leq c'' \cdot dH\sqrt{\chi}\sqrt{\phi(s,a)^\top \left(\Lambda_h^k\right)^{-1}\phi(s,a)},$$

for an absolute constant $c''$ independent of $c_{\beta'}$. Finally, to prove this lemma, we only need to show that there exists a choice of absolute constant $c_\beta$ so that

$$c''\sqrt{\iota + \log\left(c_{\beta'}+1\right)} \leq c_{\beta'}\sqrt{\iota} \tag{34}$$

where $\iota = \log(2dT/p)$. We know $\iota \in [\log 2, \infty)$ by its definition, and $c'$ is an absolute constant independent of $c_{\beta'}$. Therefore, we can pick an absolute constant $c_\beta$ which satisfies $c'\sqrt{\log 2 + \log\left(c_{\beta'}+1\right)} \leq c_{\beta'}\sqrt{\log 2}$. This choice of $c_{\beta'}$ will make Equation 34 hold for all $\iota \in [\log 2, \infty)$, which finishes the proof.

$\square$

**Lemma B.10.** *(Upper Confidence Bound (Jin et al., 2020)) Under the setting of Theorem 5.3 on the event $\mathcal{E}$ defined in Lemma B.6 we have $\hat{Q}_h^k(s,a) \geq Q_h^\star(s,a)$ for all $(s,a,h,k) \in \mathcal{S}\times\mathcal{A}\times[H]\times[K]$.*

*Proof.* We prove this lemma by induction.

First, we prove the base case, at the last step $H$. The statement holds because $\hat{Q}_H^k(s,a) \geq Q_H^\star(s,a)$. Since the value function at $H+1$ step is zero, by Lemma B.8 we have:

$$\left|\left\langle \phi(s,a), \hat{\mathbf{w}}_H^k\right\rangle - Q_H^\star(s,a)\right| \leq \beta\sqrt{\phi(s,a)^\top \left(\Lambda_H^k\right)^{-1}\phi(s,a)}.$$

Therefore, we know:

$$Q_H^\star(s,a) \leq \min \left\{ \left\langle \phi(s,a), \hat{\mathbf{w}}_H^k \right\rangle + \beta \sqrt{\phi(s,a)^\top \left( \Lambda_H^k \right)^{-1} \phi(s,a)}, H \right\} = Q_H^k(s,a).$$

Now, suppose the statement holds true at step $h+1$ and consider step $h$. Again, by LemmaB.4, we have:

$$\left| \left\langle \phi(s,a), \hat{\mathbf{w}}_h^k \right\rangle - Q_h^\star(s,a) - \mathbb{P}_h \left( \hat{V}_{h+1}^k - V_{h+1}^\star \right)(s,a) \right| \leq \beta \sqrt{\phi(s,a)^\top \left( \Lambda_h^k \right)^{-1} \phi(s,a)}.$$

By the induction assumption that $\mathbb{P}_h \left( \hat{V}_{h+1}^k - V_{h+1}^\star \right)(s,a) \geq 0$, we have:

$$Q_h^\star(s,a) \leq \min \left\{ \left\langle \phi(s,a), \hat{\mathbf{w}}_h^k \right\rangle + \beta \sqrt{\phi(s,a)^\top \left( \Lambda_h^k \right)^{-1} \phi(s,a)}, H \right\} = \hat{Q}_h^k(s,a),$$

which concludes the proof. $\qquad\square$

We will also be needing the following lemma, for lower bounding the value of our output policy $arg\max_{a \in \mathcal{A}} Q(s, \cdot)$. The following lemma shows that the pessimistic value function always lower bounds any policy value function.

**Lemma B.11.** *(Lower Confidence Bound) Under the setting of Theorem 5.3 on the event $\mathcal{E}$ defined in Lemma B.7 we have, for any policy $\pi$, $\check{Q}_h^k(s,a) \leq Q_h^\pi(s,a)$ for all $(s,a,h,k) \in \mathcal{S} \times \mathcal{A} \times [H] \times [K]$.*

*Proof.* We prove this lemma by induction similar to we just did in Lemma B.10.

Consider a fixed, arbitrary policy $\pi$, first, we prove the base case, at the last step $H$. The statement holds because $Q_H^\pi(s,a) \geq \check{Q}_H^k(s,a)$. Since the value function at $H+1$ step is zero, by Lemma B.9 we have:

$$\left| \left\langle \phi(s,a), \check{\mathbf{w}}_H^k \right\rangle - Q_H^\pi(s,a) \right| \leq \beta' \sqrt{\phi(s,a)^\top \left( \Lambda_H^k \right)^{-1} \phi(s,a)}.$$

Therefore, we know:

$$Q_H^\pi(s,a) \geq \text{clip}\left( \left\langle \phi(s,a), \check{\mathbf{w}}_H^k \right\rangle - \beta' \sqrt{\phi(s,a)^\top \left( \Lambda_H^k \right)^{-1} \phi(s,a)} \right) = \check{Q}_H^k(s,a).$$

Now, suppose the statement holds true at step $h+1$ and consider step $h$. Again, by Lemma B.9, we have:

$$\left| \left\langle \phi(s,a), \check{\mathbf{w}}_h^k \right\rangle - Q_h^\pi(s,a) - \mathbb{P}_h \left( \check{V}_{h+1}^k - V_{h+1}^\pi \right)(s,a) \right| \leq \beta' \sqrt{\phi(s,a)^\top \left( \Lambda_h^k \right)^{-1} \phi(s,a)}.$$

By the induction assumption that $\mathbb{P}_h \left( \check{V}_{h+1}^k - V_{h+1}^\pi \right)(s,a) \leq 0$, we have:

$$Q_h^\pi(s,a) \geq \text{clip}\left( \left\langle \phi(s,a), \check{\mathbf{w}}_h^k \right\rangle - \beta' \sqrt{\phi(s,a)^\top \left( \Lambda_h^k \right)^{-1} \phi(s,a)}, 0, H \right) = \check{Q}_h^k(s,a),$$

which concludes the proof. $\qquad\square$

**Theorem B.12.** *(Pseudo Regret Bound) Under Assumption B.1, for any fixed constant $\delta \in (0,1)$, with proper choice of $c > 0$, and if we set $\lambda = 1$, $\beta = c \cdot dH\sqrt{\log(2dT/\delta)}$, then with probability at least $1 - \delta$, the regret of interest of algorithm 2, $\mathbb{E}\left[ \sum_{k=1}^K V_1^\star(s_1^k) - V_1^{\pi_k}(s_1^k) \right]$, is at most $\tilde{\mathcal{O}}\left( \sqrt{d^3 H^3 T} \right)$, where $p$ is the parameter of geometric distribution.*

*Proof.* For simplicity, we use the notation:

$$\hat{\pi}_h^k(s, \cdot) = \arg\max_{a \in \mathcal{A}} \hat{Q}_h^k(s, \cdot) \quad \pi_h^k(s, \cdot) = \arg\max_{a \in \mathcal{A}} Q_h^k(s, \cdot)$$

We also denote $\mathbb{I} = \{k \in [K], L^k = 0\}$, an index set of episodes in which the trajectory is fully exploratory, then we have,

$$\mathbb{E}\left[\sum_{k=1}^K V_1^\star(s_1^k) - V_1^{\pi_k}(s_1^k)\right] \le \mathbb{E}\left[\sum_{k=1}^K \hat{V}_1^k(s_1^k) - \check{V}_1^k(s_1^k)\right] \tag{35}$$

$$= \mathbb{E}\left[\sum_{k=1}^K \hat{V}_1^{\lfloor k \rfloor}(s_1^k) - \check{V}_1^{\lfloor k \rfloor}(s_1^k)\right] \tag{36}$$

$$= \frac{1}{p} \cdot \sum_{k \in \mathbb{I}} \hat{V}_1^k(s_1^k) - \check{V}_1^k(s_1^k) \tag{37}$$

where the first step is the direct result of Lemmas B.10 and B.11, the second and the third steps are due to the construction of our algorithm, where we do not update weights until a full exploratory episode happens, and the expected interval of such event happening is $\frac{1}{p}$. And further,

$$\sum_{k \in \mathbb{I}} \hat{V}_1^k(s_1^k) - \check{V}_1^k(s_1^k) = \sum_{k \in \mathbb{I}} \hat{Q}_1^k(s_1^k, a_1^k) - \check{Q}_1^k(s_1^k, a_1'^k) \tag{38}$$

$$\le \sum_{k \in \mathbb{I}} \hat{Q}_1^k(s_1^k, a_1^k) - \check{Q}_1^k(s_1^k, a_1^k) \tag{39}$$

$$= \sum_{k \in \mathbb{I}} \left\{\Delta_h^k(s_1^k, a_1^k) - \tilde{\Delta}_h^k(s_1^k, a_1^k) + \mathbb{E}\left[\hat{V}_2^k(s_2^k) - \check{V}_2^k(s_2^k)|s_1^k, a_1^k\right]\right\} \tag{40}$$

$$\le \sum_{k \in \mathbb{I}} \left\{\beta\sqrt{\phi(s_1^k, a_1^k)^\top \left(\Lambda_h^k\right)^{-1} \phi(s_1^k, a_1^k)} + \beta'\sqrt{\phi(s_1^k, a_1^k)^\top \left(\Lambda_h^k\right)^{-1} \phi(s_1^k, a_1^k)}\right.$$

$$\left. + \mathbb{E}\left[\hat{V}_2^k(s_2^k) - \check{V}_2^k(s_2^k)|s_1^k, a_1^k\right]\right\} \tag{41}$$

$$= \sum_{k \in \mathbb{I}} \left\{\underbrace{\beta\sqrt{\phi(s_1^k, a_1^k)^\top \left(\Lambda_h^k\right)^{-1} \phi(s_1^k, a_1^k)}}_{b_1^k} + \underbrace{\beta'\sqrt{\phi(s_1^k, a_1^k)^\top \left(\Lambda_h^k\right)^{-1} \phi(s_1^k, a_1^k)}}_{b_1'^k}\right.$$

$$\tag{42}$$

$$\left. + \underbrace{\mathbb{E}\left[\hat{V}_2^k(s_2^k) - \check{V}_2^k(s_2^k)|s_1^k, a_1^k\right] - (\hat{V}_2^k(s_2^k) - \check{V}_2^k(s_2^k))}_{\zeta_2^k} + (\hat{V}_2^k(s_2^k) - \check{V}_2^k(s_2^k))\right\}$$

$$\tag{43}$$

$$= \sum_{k \in \mathbb{I}} \left[\hat{V}_2^k(s_2^k) - \check{V}_2^k(s_2^k) + b_1^k + b_1'^k + \zeta_2^k\right] \tag{44}$$

where, $a \in \arg\max_{a \in \mathcal{A}} \hat{Q}_1^k(s_1^k, \cdot)$ and $a' \in \arg\max_{a' \in \mathcal{A}} \check{Q}_1^k(s_1^k, \cdot)$.

By recursively applying Equation. (38), we have,

$$\sum_{k \in \mathbb{I}} \hat{V}_1^k(s_1^k) - \check{V}_1^k(s_1^k) \leq \sum_{k \in \mathbb{I}} \sum_{h=1}^{H} b_h^k + \sum_{k \in \mathbb{I}} \sum_{h=1}^{H} b_h'^k + \sum_{k \in \mathbb{I}} \sum_{h=1}^{H} \zeta_h^k \tag{45}$$

$$\tag{46}$$

We now bound each terms, for the first term in Equation (45), by Lemma C.2 and C.3:

$$\sum_{k \in \mathbb{I}} \sum_{h=1}^{H} b_h^k = \sum_{k \in \mathbb{I}} \sum_{h=1}^{H} \beta \sqrt{\phi(s_h^k, a_h^k)^\top \left(\Lambda_h^k\right)^{-1} \phi(s_h^k, a_h^k)} \tag{47}$$

$$\leq \sum_{h=1}^{H} \sqrt{Kp} \cdot \left[ \sum_{k \in \mathbb{I}} \beta \sqrt{\phi(s_h^k, a_h^k)^\top \left(\Lambda_h^k\right)^{-1} \phi(s_h^k, a_h^k)} \right] \tag{48}$$

$$\leq \beta \sqrt{Kp} \sum_{h=1}^{H} \sqrt{2 \log \left[ \frac{\det(\Lambda_h^k)}{\det(\Lambda_h^1)} \right]} \tag{49}$$

$$\leq \beta \sqrt{Kp} \sum_{h=1}^{H} \sqrt{2d \log \left[ \frac{\lambda + k}{\lambda} \right]} \tag{50}$$

$$\leq H \beta \iota \sqrt{2dKp} \tag{51}$$

where, the second step follows from Cauchy–Schwarz inequality, the third step follows from the Lemma C.2 and C.3, and the second last step follows from the fact that $\|\phi(\cdot, \cdot)\| \leq 1$, and thus $\|\Lambda_h^k\| \leq \lambda + k$. And following the same logic, we have, for the second term in Equation (45):

$$\sum_{k \in \mathbb{I}} \sum_{h=1}^{H} b_h'^k \leq H \beta' \iota \sqrt{2dKp}$$

For the third term in Eq(45), we notice it is a martingale difference sequence, and by applying Azuma-Hoeffding inequality, with probability at least $1 - \frac{\delta}{2}$:

$$\sum_{k \in \mathbb{I}} \sum_{h=1}^{H} \zeta_h^k \leq \sqrt{2KH^3 \log(2/\delta)} \leq 2H\sqrt{KH\iota}$$

By combining the upper of three terms in Equation (45), recall that $\beta = c \cdot dH\sqrt{log(2dT/\delta)}, \beta' = c' \cdot dH\sqrt{log(2dT/\delta)}$ we obtain:

$$\sum_{k \in \mathbb{I}} \hat{V}_1^k(s_1^k) - \check{V}_1^k(s_1^k) \leq H\beta\iota\sqrt{Kp} + H\beta'\iota\sqrt{Kp} + 2H\sqrt{KH\iota} = C' \cdot \sqrt{d^3 H^3 T \iota^2}$$

for some absolute constant $C'$.

This concludes that the total pseudo regret of policy $\pi$ over $K$ episode is given by $\tilde{\mathcal{O}}(\frac{\sqrt{d^3 H^3 T \iota^2}}{p})$. And equivalently, we conclude that our algorithm obtains $\epsilon - optimal$ policy with $\tilde{\mathcal{O}}(\frac{d^3 H^4}{p\epsilon^2})$ samples with probability at least $1 - \delta$.

$\square$

## C  Auxiliary Lemmas

**Lemma C.1.** *(Jin et al., 2020) Let $\Lambda_t = \lambda \mathbf{I} + \sum_{i=1}^{t} \phi_i \phi_i^\top$ where $\phi_i \in \mathbb{R}^d$ and $\lambda > 0$. Then:*

$$\sum_{i=1}^{t} \phi_i^\top \left(\Lambda_t\right)^{-1} \phi_i \le d$$

*Proof.* We have $\sum_{i=1}^{t} \phi_i^\top \left(\Lambda_t\right)^{-1} \phi_i = \sum_{i=1}^{t} \operatorname{tr}\left(\phi_i^\top \left(\Lambda_t\right)^{-1} \phi_i\right) = \operatorname{tr}\left(\left(\Lambda_t\right)^{-1} \sum_{i=1}^{t} \phi_i \phi_i^\top\right)$. Given the eigenvalue decomposition $\sum_{i=1}^{t} \phi_i \phi_i^\top = \mathbf{U} \operatorname{diag}\left(\lambda_1, \ldots, \lambda_d\right) \mathbf{U}^\top$, we have $\Lambda_t = \mathbf{U} \operatorname{diag}\left(\lambda_1 + \lambda, \ldots, \lambda_d + \lambda\right) \mathbf{U}^\top$, and $\operatorname{tr}\left(\left(\Lambda_t\right)^{-1} \sum_{i=1}^{t} \phi_i \phi_i^\top\right) = \sum_{j=1}^{d} \lambda_j / \left(\lambda_j + \lambda\right) \le d$.content... $\square$

**Lemma C.2.** *(Abbasi-Yadkori et al., 2011) Let $\{\phi_t\}_{t \ge 0}$ be a bounded sequence in $\mathbb{R}^d$ satisfying $\sup_{t \ge 0} \|\phi_t\| \le 1$. Let $\Lambda_0 \in \mathbb{R}^{d \times d}$ be a positive definite matrix. For any $t \ge 0$, we define $\Lambda_t = \Lambda_0 + \sum_{j=1}^{t} \phi_j \phi_j^\top$. Then, if the smallest eigenvalue of $\Lambda_0$ satisfies $\lambda_{\min}\left(\Lambda_0\right) \ge 1$, we have*

$$\log\left[\frac{\det\left(\Lambda_t\right)}{\det\left(\Lambda_0\right)}\right] \le \sum_{j=1}^{t} \phi_j^\top \Lambda_{j-1}^{-1} \phi_j \le 2\log\left[\frac{\det\left(\Lambda_t\right)}{\det\left(\Lambda_0\right)}\right]$$

*Proof.* Since $\lambda_{\min}\left(\Lambda_0\right) \ge 1$ and $\|\phi_t\| \le 1$ for all $j \ge 0$, we have

$$\phi_j^\top \Lambda_{j-1}^{-1} \phi_j \le \left[\lambda_{\min}\left(\Lambda_0\right)\right]^{-1} \cdot \|\phi_j\|^2 \le 1, \quad \forall j \ge 0.$$

Note that, for any $x \in [0, 1]$, it holds that $\log(1 + x) \le x \le 2\log(1 + x)$. Therefore, we have

$$\sum_{j=1}^{t} \log\left(1 + \phi_j^\top \Lambda_{j-1}^{-1} \phi_j\right) \le \sum_{j=1}^{t} \phi_j^\top \Lambda_{j-1}^{-1} \phi_j \le 2\sum_{j=1}^{t} \log\left(1 + \phi_j^\top \Lambda_{j-1}^{-1} \phi_j\right) \tag{52}$$

Moreover, for any $t \ge 0$, by the definition of $\Lambda_t$, we have

$$\det\left(\Lambda_t\right) = \det\left(\Lambda_{t-1} + \phi_t \phi_t^\top\right) = \det\left(\Lambda_{t-1}\right) \cdot \det\left(\mathbf{I} + \Lambda_{t-1}^{-1/2} \phi_t \phi_t^\top \Lambda_{t-1}^{-1/2}\right)$$

Since $\det\left(\mathbf{I} + \Lambda_{t-1}^{-1/2} \phi_t \phi_t^\top \Lambda_{t-1}^{-1/2}\right) = 1 + \phi_t^\top \Lambda_{t-1}^{-1} \phi_t$, the recursion gives:

$$\sum_{j=1}^{t} \log\left(1 + \phi_j^\top \Lambda_{j-1}^{-1} \phi_j\right) = \log\det\left(\Lambda_t\right) - \log\det\left(\Lambda_0\right) \tag{53}$$

Therefore, combining Equation (52) and Equation (53), we conclude the proof. $\square$

In our algorithm, full-exploratory trajectory occasionally occurs, and other trajectories also contributes our parameter $\Lambda_h^k$, in the following Lemma, we show that by adding more data, the bound remains effective.

**Lemma C.3.** *Let $\{\phi_t\}_{t \ge 0}$ be a bounded sequence in $\mathbb{R}^d$ satisfying $\sup_{t \ge 0} \|\phi_t\| \le 1$. And let $\{\psi_s\}_{s \ge 0}$ be another sequence of in $\mathbb{R}^d$ satisfying $\sup_{s \ge 0} \|\psi_s\| \le 1$. Let $\Lambda_0 \in \mathbb{R}^{d \times d}$ be a positive definite matrix. For any $t \ge 0$, $s \ge 0$, we define $\Lambda_t = \Lambda_0 + \sum_{j=1}^{t} \phi_j \phi_j^\top$, $\Lambda_{t,s} = \Lambda_0 + \sum_{j=1}^{t} \phi_j \phi_j^\top + \sum_{i=1}^{s} \psi_i \psi_i^\top$. Then, if the smallest eigenvalue of $\Lambda_0$ satisfies $\lambda_{\min}\left(\Lambda_0\right) \ge 1$, we have*

$$\sum_{j=1}^{t} \phi_j^\top \Lambda_{j-1,s_j}^{-1} \phi_j \le 2\log\left[\frac{\det\left(\Lambda_t\right)}{\det\left(\Lambda_0\right)}\right]$$

*where $\{s_j\}_{1 \le j \le t}$ is any non-decreasing sequence of number satisfying $s_j \in \mathbb{N}$.*

*Proof.* Consider any $t, s \in \mathbb{N}$, since $\Lambda_0$ is positive definite, and $\sum_{j=1}^{t} \phi_j \phi_j^\top$ and $\sum_{i=1}^{s} \psi_i \psi_i^\top$ are semi-positive-definite, we know that $\sigma(\Lambda_{t,s}) \geq \sigma(\Lambda_t)$ and $\sigma(\Lambda_t^{-1}) \geq \sigma(\Lambda_{t,s}^{-1})$ in a pointwise manner. This gives us, for any sequence $\{s_j\}_{1 \leq j \leq t}, s_j \in \mathbb{N}$,

$$\sum_{j=1}^{t} \phi_j^\top \Lambda_{j-1,s_j}^{-1} \phi_j \leq \sum_{j=1}^{t} \phi_j^\top \Lambda_{j-1}^{-1} \phi_j \leq 2 \log \left[ \frac{\det(\Lambda_t)}{\det(\Lambda_0)} \right]$$

This concludes the proof.

$\square$

**Lemma C.4.** *(Concentration of Self-Normalized Processes (Abbasi-Yadkori et al., 2011)). Let $\{\varepsilon_t\}_{t=1}^{\infty}$ be a real-valued stochastic process with corresponding filtration $\{\mathcal{F}_t\}_{t=0}^{\infty}$. Let $\varepsilon_t \mid \mathcal{F}_{t-1}$ be zero-mean and $\sigma$-subGaussian; i.e. $\mathbb{E}[\varepsilon_t \mid \mathcal{F}_{t-1}] = 0$, and*

$$\forall \lambda \in \mathbb{R}, \quad \mathbb{E}\left[ e^{\lambda \varepsilon_t} \mid \mathcal{F}_{t-1} \right] \leq e^{\lambda^2 \sigma^2 / 2}.$$

*Let $\{_t\}_{t=0}^{\infty}$ be an $\mathbb{R}^d$-valued stochastic process where $\phi_t \in \mathcal{F}_{t-1}$. Assume $\Lambda_0$ is a $d \times d$ positive definite matrix, and let $\Lambda_t = \Lambda_0 + \sum_{s=1}^{t} \phi_s \phi_s^\top$. Then for any $\delta > 0$, with probability at least $1 - \delta$, we have for all $t \geq 0$ :*

$$\left\| \sum_{s=1}^{t} \phi_s \varepsilon_s \right\|_{\Lambda_t^{-1}}^2 \leq 2\sigma^2 \log \left[ \frac{\det(\Lambda_t)^{1/2} \det(\Lambda_0)^{-1/2}}{\delta} \right]$$

**Lemma C.5.** *(Jin et al., 2020) Let $\{s_\tau\}_{\tau=1}^{\infty}$ be a stochastic process on state space $\mathcal{S}$ with corresponding filtration $\{\mathcal{F}_\tau\}_{\tau=0}^{\infty}$. Let $\{\phi_\tau\}_{\tau=0}^{\infty}$ be an $\mathbb{R}^d$-valued stochastic process where $\phi_\tau \in \mathcal{F}_{\tau-1}$, and $\|\phi_\tau\| \leq 1$. Let $\Lambda_k = \lambda I + \sum_{\tau=1}^{k} \phi_\tau \phi_\tau^\top$. Then for any $\delta > 0$, with probability at least $1 - \delta$, for all $k \geq 0$, and any $V \in \mathcal{V}$ so that $\sup_s |V(s)| \leq H$, we have:*

$$\left\| \sum_{\tau=1}^{k} \phi_\tau \{V(s_\tau) - \mathbb{E}[V(s_\tau) \mid \mathcal{F}_{\tau-1}]\} \right\|_{\Lambda_k^{-1}}^2 \leq 4H^2 \left[ \frac{d}{2} \log \left( \frac{k+\lambda}{\lambda} \right) + \log \frac{\mathcal{N}_\varepsilon}{\delta} \right] + \frac{8k^2 \varepsilon^2}{\lambda},$$

*where $\mathcal{N}_\varepsilon$ is the $\varepsilon$-covering number of $\mathcal{V}$ with respect to the distance $\mathrm{dist}(V, V') = \sup_s |V(s) - V'(s)|$.*

*Proof.* For any $V \in \mathcal{V}$, we know there exists a $\tilde{V}$ in the $\varepsilon$-covering such that

$$V = \widetilde{V} + \Delta_V \quad \text{and} \quad \sup_s |\Delta_V(s)| \leq \varepsilon$$

This gives following decomposition:

$$\left\| \sum_{\tau=1}^{k} \phi_\tau \{V(s_\tau) - \mathbb{E}[V(s_\tau) \mid \mathcal{F}_{\tau-1}]\} \right\|_{\Lambda_k^{-1}}^2 \tag{54}$$

$$\leq 2 \left\| \sum_{\tau=1}^{k} \phi_\tau \left\{ \widetilde{V}(s_\tau) - \mathbb{E}\left[\widetilde{V}(s_\tau) \mid \mathcal{F}_{\tau-1}\right] \right\} \right\|_{\Lambda_k^{-1}}^2 + 2 \left\| \sum_{\tau=1}^{k} \phi_\tau \{\Delta_V(s_\tau) - \mathbb{E}[\Delta_V(s_\tau) \mid \mathcal{F}_{\tau-1}]\} \right\|_{\Lambda_k^{-1}}^2, \tag{55}$$

where we can apply Theorem D.3 and a union bound to the first term. Also, it is not hard to bound the second term by $8k^2 \varepsilon^2 / \lambda$.

To compute the covering number of function class $\mathcal{V}$, we first require a basic result on the covering number of a Euclidean ball as follows. We refer readers to classical material, such as Lemma 5.2 in [44], for its proof. Lemma D.5 (Covering Number of Euclidean Ball). For any $\varepsilon > 0$, the $\varepsilon$-covering number of the Euclidean ball in $\mathbb{R}^d$ with radius $R > 0$ is upper bounded by $(1 + 2R/\varepsilon)^d$. $\qquad\square$

**Lemma C.6.** *(Covering Number of Euclidean Ball). For any $\varepsilon > 0$, the $\varepsilon$-covering number of the Euclidean ball in $\mathbb{R}^d$ with radius $R > 0$ is upper bounded by $(1 + 2R/\varepsilon)^d$.*

Based on the lemmas above, we can bound the covering number of the optimistic value function and pessimistic value function class.

**Lemma C.7.** *(Covering number of optimistic function class (Jin et al., 2020)) Let $\mathcal{V}$ denote a class of functions mapping from $\mathcal{S}$ to $\mathbb{R}$ with following parametric form*

$$V(\cdot) = \min\left\{\max_a \mathbf{w}^\top \boldsymbol{\phi}(\cdot, a) + \beta\sqrt{\boldsymbol{\phi}(\cdot, a)^\top \Lambda^{-1} \boldsymbol{\phi}(\cdot, a)}, H\right\}$$

*where the parameters $(\mathbf{w}, \beta, \Lambda)$ satisfy $\|\mathbf{w}\| \leq L, \beta \in [0, B]$ and the minimum eigenvalue satisfies $\lambda_{\min}(\Lambda) \geq \lambda$. Assume $\|\boldsymbol{\phi}(s, a)\| \leq 1$ for all $(s, a)$ pairs, and let $\mathcal{N}_\varepsilon$ be the $\varepsilon$-covering number of $\mathcal{V}$ with respect to the distance $\mathrm{dist}(V, V') = \sup_s |V(s) - V'(s)|$. Then*

$$\log \mathcal{N}_\varepsilon \leq d\log(1 + 4L/\varepsilon) + d^2 \log\left[1 + 8d^{1/2}B^2/\left(\lambda\varepsilon^2\right)\right]$$

*Proof.* Equivalently, we can reparametrize the function class $\mathcal{V}$ by let $\mathbf{A} = \beta^2 \Lambda^{-1}$, so we have

$$V(\cdot) = \min\left\{\max_a \mathbf{w}^\top \boldsymbol{\phi}(\cdot, a) + \sqrt{\boldsymbol{\phi}(\cdot, a)^\top \mathbf{A} \boldsymbol{\phi}(\cdot, a)}, H\right\} \tag{56}$$

for $\|\mathbf{w}\| \leq L$ and $\|\mathbf{A}\| \leq B^2\lambda^{-1}$. For any two functions $V_1, V_2 \in \mathcal{V}$, let them take the form in Equation (56) with parameters $(\mathbf{w}_1, \mathbf{A}_1)$ and $(\mathbf{w}_2, \mathbf{A}_2)$, respectively. Then, since both $\min\{\cdot, H\}$ and $\max_a$ are contraction maps, we have

$$\mathrm{dist}(V_1, V_2) \leq \sup_{s,a} \left|\left[\mathbf{w}_1^\top \boldsymbol{\phi}(s, a) + \sqrt{\boldsymbol{\phi}(s, a)^\top \mathbf{A}_2 \boldsymbol{\phi}(s, a)}\right] - \left[\mathbf{w}_2^\top \boldsymbol{\phi}(s, a) + \sqrt{\boldsymbol{\phi}(s, a)^\top \mathbf{A}_2 \boldsymbol{\phi}(s, a)}\right]\right| \tag{57}$$

$$\leq \sup_{\boldsymbol{\phi}:\|\boldsymbol{\phi}\|\leq 1} \left|\left[\mathbf{w}_1^\top \boldsymbol{\phi} + \sqrt{\boldsymbol{\phi}^\top \mathbf{A}_2 \boldsymbol{\phi}}\right] - \left[\mathbf{w}_2^\top \boldsymbol{\phi} + \sqrt{\boldsymbol{\phi}^\top \mathbf{A}_2 \boldsymbol{\phi}}\right]\right| \tag{58}$$

$$\leq \sup_{\boldsymbol{\phi}:\|\boldsymbol{\phi}\|\leq 1} \left|(\mathbf{w}_1 - \mathbf{w}_2)^\top \boldsymbol{\phi}\right| + \sup_{\boldsymbol{\phi}:\|\boldsymbol{\phi}\|\leq 1} \sqrt{\left|\boldsymbol{\phi}^\top (\mathbf{A}_1 - \mathbf{A}_2) \boldsymbol{\phi}\right|} \tag{59}$$

$$= \|\mathbf{w}_1 - \mathbf{w}_2\| + \sqrt{\|\mathbf{A}_1 - \mathbf{A}_2\|} \leq \|\mathbf{w}_1 - \mathbf{w}_2\| + \sqrt{\|\mathbf{A}_1 - \mathbf{A}_2\|_F} \tag{60}$$

where the second last inequality follows from the fact that $|\sqrt{x} - \sqrt{y}| \leq \sqrt{|x - y|}$ holds for any $x, y \geq 0$. For matrices, $\|\cdot\|$ and $\|\cdot\|_F$ denote the matrix operator norm and Frobenius norm respectively.

Let $\mathcal{C}_\mathbf{w}$ be an $\varepsilon/2$-cover of $\{\mathbf{w} \in \mathbb{R}^d \mid \|\mathbf{w}\| \leq L\}$ with respect to the 2-norm, and $\mathcal{C}_\mathbf{A}$ be an $\varepsilon^2/4$-cover of $\{\mathbf{A} \in \mathbb{R}^{d\times d} \mid \|\mathbf{A}\|_F \leq d^{1/2}B^2\lambda^{-1}\}$ with respect to the Frobenius norm. By Lemma C.6, we know:

$$|\mathcal{C}_\mathbf{w}| \leq (1 + 4L/\varepsilon)^d, \quad |\mathcal{C}_\mathbf{A}| \leq \left[1 + 8d^{1/2}B^2/\left(\lambda\varepsilon^2\right)\right]^{d^2}$$

By Equation (57), for any $V_1 \in \mathcal{V}$, there exists $\mathbf{w}_2 \in \mathcal{C}_\mathbf{w}$ and $\mathbf{A}_2 \in \mathcal{C}_\mathbf{A}$ such that $V_2$ parametrized by $(\mathbf{w}_2, \mathbf{A}_2)$ satisfies $\mathrm{dist}(V_1, V_2) \leq \varepsilon$. Hence, it holds that $\mathcal{N}_\varepsilon \leq |\mathcal{C}_\mathbf{w}| \cdot |\mathcal{C}_\mathbf{A}|$, which gives:

$$\log \mathcal{N}_\varepsilon \le \log |\mathcal{C}_\mathbf{w}| + \log |\mathcal{C}_\mathbf{A}| \le d \log(1 + 4L/\varepsilon) + d^2 \log \left[ 1 + 8d^{1/2}B^2 / \left(\lambda \varepsilon^2\right) \right]$$

This concludes the proof. $\qquad\square$

And we can obtain the same covering number bound on our pessimistic value function class due to the symmetry.

**Lemma C.8.** *(Covering number of pessimistic function class) Let $\mathcal{V}$ denote a class of functions mapping from $\mathcal{S}$ to $\mathbb{R}$ with following parametric form*

$$V(\cdot) = \text{clip} \left( \max_a \mathbf{w}^\top \boldsymbol{\phi}(\cdot, a) - \beta \sqrt{\boldsymbol{\phi}(\cdot, a)^\top \Lambda^{-1} \boldsymbol{\phi}(\cdot, a)}, 0, H \right)$$

*where the parameters $(\mathbf{w}, \beta, \Lambda)$ satisfy $\|\mathbf{w}\| \le L, \beta \in [0, B]$ and the minimum eigenvalue satisfies $\lambda_{\min}(\Lambda) \ge \lambda$. Assume $\|\boldsymbol{\phi}(s, a)\| \le 1$ for all $(s, a)$ pairs, and let $\mathcal{N}_\varepsilon$ be the $\varepsilon$-covering number of $\mathcal{V}$ with respect to the distance $\text{dist}\,(V, V') = \sup_s |V(s) - V'(s)|$. Then*

$$\log \mathcal{N}_\varepsilon \le d \log(1 + 4L/\varepsilon) + d^2 \log \left[ 1 + 8d^{1/2}B^2 / \left(\lambda \varepsilon^2\right) \right]$$

*Proof.* Similar to the proof strategy used in Lemma C.7, we reparametrize the function the function class $\mathcal{V}$ by letting $\mathbf{A} = \beta^2 \Lambda^{-1}$, which gives us,

$$V(\cdot) = \text{clip} \left( \max_a \mathbf{w}^\top \boldsymbol{\phi}(\cdot, a) - \sqrt{\boldsymbol{\phi}(\cdot, a)^\top \mathbf{A} \boldsymbol{\phi}(\cdot, a)}, 0, H \right) \qquad (61)$$

for $\|\mathbf{w}\| \le L$ and $\|\mathbf{A}\| \le B^2 \lambda^{-1}$. For any two functions $V_1, V_2 \in \mathcal{V}$, let them take the form in Equation (61) with parameters $(\mathbf{w}_1, \mathbf{A}_1)$ and $(\mathbf{w}_2, \mathbf{A}_2)$, respectively. Then, since both $\text{clip}(\cdot, 0, H)$ and $\max_a$ are contraction maps, we have

$$\text{dist}\,(V_1, V_2) \le \sup_{s,a} \left| \left[ \mathbf{w}_1^\top \boldsymbol{\phi}(s, a) - \sqrt{\boldsymbol{\phi}(s, a)^\top \mathbf{A}_1 \boldsymbol{\phi}(s, a)} \right] - \left[ \mathbf{w}_2^\top \boldsymbol{\phi}(s, a) - \sqrt{\boldsymbol{\phi}(s, a)^\top \mathbf{A}_2 \boldsymbol{\phi}(s, a)} \right] \right| \qquad (62)$$

$$\le \sup_{\boldsymbol{\phi}: \|\boldsymbol{\phi}\| \le 1} \left| \left[ \mathbf{w}_1^\top \boldsymbol{\phi} - \sqrt{\boldsymbol{\phi}^\top \mathbf{A}_1 \boldsymbol{\phi}} \right] - \left[ \mathbf{w}_2^\top \boldsymbol{\phi} - \sqrt{\boldsymbol{\phi}^\top \mathbf{A}_2 \boldsymbol{\phi}} \right] \right| \qquad (63)$$

$$\le \sup_{\boldsymbol{\phi}: \|\boldsymbol{\phi}\| \le 1} \left| (\mathbf{w}_1 - \mathbf{w}_2)^\top \boldsymbol{\phi} \right| + \sup_{\boldsymbol{\phi}: \|\boldsymbol{\phi}\| \le 1} \sqrt{\left| \boldsymbol{\phi}^\top (\mathbf{A}_2 - \mathbf{A}_1) \boldsymbol{\phi} \right|} \qquad (64)$$

$$= \|\mathbf{w}_1 - \mathbf{w}_2\| + \sqrt{\|\mathbf{A}_2 - \mathbf{A}_1\|} \le \|\mathbf{w}_1 - \mathbf{w}_2\| + \sqrt{\|\mathbf{A}_2 - \mathbf{A}_1\|_F} \qquad (65)$$

$$= \|\mathbf{w}_1 - \mathbf{w}_2\| + \sqrt{\|\mathbf{A}_1 - \mathbf{A}_2\|_F} \qquad (66)$$

Equation (62) shows that the distance of two elements in pessimistic value function class shares a same upper bound with the optimistic value function class. By Lemma C.7, we conclude the proof. $\qquad\square$

