# OpenReview forum: "DEXR: A Unified Approach Towards Environment Agnostic Exploration"
_ICLR.cc/2024/Conference — Submitted to ICLR 2024_

### Official Review · Reviewer_NQDk · 2023-10-23

**Soundness:** 4 excellent
**Presentation:** 4 excellent
**Contribution:** 2 fair
**Rating:** 5
**Confidence:** 5

**Summary:**

The paper proposes a framework for exploration in which the task policy is rolled out first and the exploration policy continues after. This is compared to related work such as rolling exploration first or optimizing a joint objective. There are experiments on continuous control mazes and mujoco tasks.

**Strengths:**

- The paper proposes a straightforward framework that intuitively should work well
- The results are promising

**Weaknesses:**

- The method is incremental compared to work like Go-Explore.
- There are missing baselines. Specifically, the method is never compared to the same method but without exploration, i.e. the TD3 baseline.

**Questions:**

Comparing the results to the original TD3 paper seems that the proposed method does not outperform TD3. Furthermore, the baseline exploration methods reported all perform worse than TD3. Why does this happen? Since these methods are implemented on top of TD3 it would be strange if they had worse performance than the backbone RL method without any extrinsic rewards.

---

> ### Author Response · Authors · 2023-11-20
>
> We want to thank the reviewer NQDk for the effort in reviewing the paper. We address the questions and respond to the concerns in the following.
>
> Q1: "Comparing the results ... does not outperform TD3."
>
> A1: In our PointMaze experiments, DEXR consistently outperforms TD3 across various layouts, achieving significantly better results. Specifically, we observed that TD3 requires at least twice as long to reach the goal point compared to DEXR. This demonstrates the superior efficiency and effectiveness of DEXR in these scenarios.
>
> Regarding the MuJoCo Locomotion tasks, our results indicate a more nuanced performance comparison. While DEXR does not show a substantial advantage over TD3 in environments with dense rewards, it notably surpasses TD3's performance in most environments characterized by sparse rewards. This distinction highlights DEXR's enhanced adaptability and proficiency in more challenging settings with limited feedback.
>
> Q2: Furthermore, the baseline exploration methods reported all perform worse than TD3. Why does this happen? Since these methods are implemented on top of TD3 it would be strange if they had worse performance than the backbone RL method without any extrinsic rewards.
>
> A2: As discussed extensively in Sections 1 and 2, methods like Exp, which rely on intrinsic rewards, tend to be distracted and exhibit unstable performance, failing to exploit efficiently when the hyperparameter is not carefully tuned. This is the precise challenge DEXR aims to address.
>
> As detailed in Section 5, all algorithms, including TD3, are trained using extrinsic rewards. Notably, the TD3 agent is trained without the addition of intrinsic rewards.
>
> Concern 1: The method is incremental compared to work like Go-Explore.
>
> Response 1: We regret any misunderstanding regarding the relationship between DEXR and Go-Explore. While there is a superficial similarity, DEXR embodies a fundamentally different design philosophy and operates within a markedly distinct setting.
>
> Firstly, Go-Explore's methodology hinges on the environment being deterministic or the availability of a local model (resettable environment) that can relocate the agent to a previously visited state. Contrarily, DEXR operates independently of these requirements.
>
> Secondly, Go-Explore relocates the agent to regions that are rarely visited previously, whereas DEXR relocates the agent to regions that are believed to be fruitful according to the extra exploitation policy. Despite these two paradigms being similar in form, DEXR and Go-Explore hold drastically different ideas and intuition. To be specific, Go-Explore enables more efficient **over-exploration**, by leveraging the special assumptions and privilege accesses to accurately relocate the agent to “regions not well covered”, so that the agent can fully cover the whole MDP, including regions that might not be worth exploring. In contrast, DEXR prevents **over-exploration** and does not need any special assumptions, by employing an extra exploitation policy to relocate the agent to regions that are “worth exploring”, which largely mitigates the hyperparameter sensitivity, as discussed in Section 1 and Section 4 in detail.
>
> The effectiveness of DEXR, while straightforward and intuitive, is underscored by our comprehensive empirical analysis. This analysis examines the behaviors of the exploitation and exploration policies in navigation and evaluates the capacity of the exploitation policy to effectively guide the exploration policy.
>
> We have already put a discussion on the relationship between DEXR and the Go-Explore algorithm in the related work section, the reviewer can refer to the revised manuscript for a clearer context.
>
> Concern 2: "There are missing baselines ... TD3 baseline."
>
> Weakness 2: We acknowledge the inquiry regarding the comparison with TD3. For your reference, this comparison can be found in the context of the PointMaze experiment. Additionally, we provide visual representations of the behavior of various algorithms, including TD3, within the PointMaze environment, detailed on page 8 of our document.
>
> We apologize for the omission of the split plots and the comparative plot against TD3 in the appendix. We have since rectified this oversight, and these plots are now included in the updated version of the manuscript and can be found in Appendix A.

---

> > ### Comment · Reviewer_NQDk · 2023-11-21
> > **The experimental evidence insufficient**
> >
> > - I thank the authors for the updated results and I increase my score accordingly. However, I find the experimental evidence for the benefits of using the algorithm insufficient, which prevents me from accepting the paper
> >   - A key claim of the paper is that DEXR is more robust to different values of the balance hyperparameter. However, setting the balance to 1 yields good results for the baselines evaluated in this paper. While the proposed method works better with much higher values, it is unclear why that is notable. Balance term of 1 seems like a reasonable default value.
> >   - With beta=1, the proposed method produces mixed results relatively to the baselines and does not definitively outperform them.
> >   - The robustness results are also mixed. DEXR's performance degrades with larger balance terms, and in the aggregate setting DEXR performs somewhat similarly to baselines on 3 out of 5 sparse reward tasks, and clearly outperforms the baselines on only 2 tasks.
> > - I regret any misunderstanding regarding Go-Explore. While the major result in the go-explore paper requires the ability to reset the agent to an arbitrary state, the paper also proposes a version of the method that employs goal-conditioned policies to reach the desired state. This is further extended in Ecoffet'20. I find the discussion of the reset assumption confusing in light of these results.
> >
> > Ecoffet'20, First return, then explore

---

> > > ### Author Response · Authors · 2023-11-22
> > >
> > > We thank the reviewer NQDk's effort in reviewing our paper, here are our responses:
> > > * $\beta=1.0$ is the most natural choice, but it is not always the reasonable default value, in fact, it is hard to identify such a "reasonable default value" for different intrinsic rewards and environments, as we extensively discussed in Section 1. If you refer to the performance plot in Appendix A, with Dynamics intrinsic reward, $\beta=1.0$ yields quite poor performance on baselines. This is because the Dynamics intrinsic reward outputs a larger value, which means the "reasonable default" value for this intrinsic reward should be smaller than 1.0.
> > > * $\beta=1.0$ is suitable for the Disagreement intrinsic reward, which enables the standard exploration method to learn efficiently. As we discussed in Section 1., when the properties of the environments and the intrinsic reward model are not known a priori, we would want a larger intrinsic reward coefficient for better exploration, but the problems of the standard algorithms prevent one from doing so because the larger coefficients cause optimization problem. We emphasize that our aim is not to show DEXR outperforms some other baseline under some specific $\beta$ but to show its robustness when $\beta$ changes.
> > > * Even with the goal-conditioned policy, DEXR still has a very different philosophy. The goal-conditioned policy-based Go-Explore algorithm still aims to reach states that are rarely visited, instead of going to high-reward regions to prevent over-exploration.
> > >
> > > We kindly refer to Section 1, Section 4, and Appendix A for the reviewer to understand the details about the motivation and the aim of our proposed algorithm.

---

### Official Review · Reviewer_CdTY · 2023-10-31

**Soundness:** 2 fair
**Presentation:** 3 good
**Contribution:** 2 fair
**Rating:** 5
**Confidence:** 3

**Summary:**

This paper proposes a new exploration scheme to reduce the sensitivity of exploration hyperparameters across different environments. The new scheme is a two-phased process, starting with a pure exploitative policy and then switching to an exploratory policy at each episode. Specifically, the exploratory policy will try to take over the episode according to a Bernoulli event at each time step. Empirical results on some navigation tasks and MuJoCo controls tasks show consistent performance improvement over baseline algorithms.

**Strengths:**

The strengths of the paper are its originality and significance. To the best of my knowledge, the idea of the paper is novel. The exploration-exploitation trade-off has been an outstanding issue in reinforcement learning, and the idea of this paper provides a simple but effective way to handle the trade-off. It’s interesting to see how an extra exploitation phase at the beginning of each episode helps deeper exploration, which may be of interest to the broad RL community.

In addition, the theoretical analysis of DEXR-UCB (LSVI-UCB with DEXR) is somewhat interesting. I suggest the author(s) lay out the condition on the intrinsic motivation reward under which DEXR is theoretically sound, which would make the theoretical results more appealing.

**Weaknesses:**

In terms of weaknesses, the paper can be improved in its soundness and quality:
1. First of all, Appendix B is still missing, which contains important empirical results to validate the paper's central claim. The paper claims to find an algorithm that has a reduced sensitivity to the exploration hyperparameter. Then, how the algorithm (and baselines) reacts to different values of the exploration hyperparameter should be shown. Also, I think it would be better if it’s in the main text.
2. The effect of the extra hyperparameter, the truncation probability $p$, is not discussed adequately. The paper only mentions that it is set to $1-\gamma$. In addition, there is an annealing of this parameter, which is not mentioned in the main text nor investigated thoroughly.  Are there any justifications for these design choices?
3. There are still a lot of typos and reference format issues in the paper. Please carefully fix them. See Questions for an unexhausted list.

I will reevaluate the paper if the authors address the above concerns.

**Questions:**

1. How does $\beta$ influence the performance of each algorithm?
2. Are there any justifications for setting $p$ to $1-\gamma$? Also, why is $p$ annealed to $\frac{1}{H}$ during the course of the training?

Typos and minor suggestions:
1. At the top of page 4, the two $\pi$s seem to be missing a superscript $^{ext}$.
2. In Eq. 3 and Eq. 4 on page 4, $V(s, a)$ should be $V(s)$.
3. It may be a good idea to also define $b$ in the input of Algorithm 1. Otherwise, the reader may be confused if they just read the pseudocode.
4. At the bottom of page 8, “for for” should be “for.”
5. In the first paragraph of page 13, a verb is missing between “then $\frac{1}{1-\gamma}.”
6. There are incorrect or inconsistent reference formats scattered throughout. Please fix them.

---

> ### Author Response · Authors · 2023-11-20
>
> We want to thank the reviewer CdTY for the effort in reviewing our paper, we address the questions and concerns in the following.
>
> Q1: How does $\beta$ influence the performance of each algorithm?
>
> A1: We apologize for the oversight concerning the absence of split plots in the appendix of our manuscript. We have already put the split results in Appendix A, which describes the influence of $\beta$ on each algorithm.
>
> Q2: Are there any justifications for setting $p$ to $1-\gamma$? Also, why is $p$ annealed to $\frac{1}{H}$ during the course of the training?
>
> A2: Let me walk through the key choices in detail:
>
> Using $p=1-\gamma$ for truncation probability initially:
>
> - The discount factor $\gamma$ determines the effective horizon $H^e=1/(1-\gamma)$
> - This horizon suggests the steps needed to propagate rewards back to early states
> - Therefore, letting the exploitation policy run for H steps on average via p=1-γ allows it to reach promising states
> - Formally, if $T \sim Geom(p)$, then $\mathbb E[T]=1/p=1/(1-γ)=H^e$
>
> Decaying $p$ over time:
>
> - As exploitation policy improves, less exploration is needed near the initial point
> - Reducing p increases $\mathbb E[T]$ since $\mathbb E[T]=1/p$
> - This lets exploitation run longer to fine-tune on higher-value regions
> - $p$ is decayed gradually to $1/H$, to gradually increase $\mathbb E[T]$ to $H$
>
> Avoiding excessive pointless exploitation:
>
> - Despite higher $\mathbb E[T]$ later, $T$ still follows a geometric distribution
> - So there are always probabilistic chances of truncating early
> - Formally, $\mathbb P(T=t)=(1-p)^{t-1}p$ still enables occasional exploration in region near the initial point
> - Experiments confirm this prevents issues of over-exploitation
>
> Per your feedback, we have already added some discussion on this design choice in Section 4 for more information.
>
> Q3: Typos and minor suggestions.
>
> A3: We extend our apologies for any inconvenience caused by these errors. Additionally, we want to clarify that in Equations 3 and 4, the term $\mathbb{P}V(s, a)$ is defined as $\mathbb{E}_{s’ \sim \mathbb{P}(\cdot|s, a)} V(s')$, as elaborated in the paragraph subsequent to Equation 2.
>
> Concern 1: First of all, Appendix B is still missing, which contains important empirical results to validate the paper's central claim. The paper claims to find an algorithm that has a reduced sensitivity to the exploration hyperparameter. Then, how the algorithm (and baselines) reacts to different values of the exploration hyperparameter should be shown. Also, I think it would be better if it’s in the main text.
>
> Response 1: We again apologize for the oversight concerning the absence of split plots in the appendix of our manuscript. To rectify this, we have now included a separate plot in Appendix A.
>
> Concern 2: The effect of the extra hyperparameter, the truncation probability, is not discussed adequately. The paper only mentions that it is set to $1-\gamma$. In addition, there is an annealing of this parameter, which is not mentioned in the main text nor investigated thoroughly. Are there any justifications for these design choices?
>
> Response 2:  We have addressed this concern in Q2, the reviewer can refer to the details in our answer A2.
>
> Concern 3: There are still a lot of typos and reference format issues in the paper. Please carefully fix them. See Questions for an unexhausted list.
>
> Response 3: We appreciate your suggestion and would like to inform you that the necessary corrections have been made.

---

> > ### Comment · Reviewer_CdTY · 2023-11-22
> >
> > Thank you for the detailed reply. The updated version addresses some of my concerns about the missing results and presentation. I've updated my assessment accordingly. Here are some follow-ups:
> > > We have already put the split results in Appendix A, which describes the influence of $\beta$
> >  on each algorithm.
> >
> > The added learning curve plots are very useful for inspecting the behavior of algorithms with specific parameter values. However, there are too many plots, and it’s not very obvious to see whether the proposed method is less sensitive to $\beta$. Maybe it's a good idea to summarize them with sensitivity plots. For example, use the x-axis to show different values of $\beta$ and the y-axis to show a corresponding performance metric.
> >
> > > This horizon suggests the steps needed to propagate rewards back to early states
> >
> > I know the expectation of the termination time step of the exploitation policy would be $1/p$. But what do you mean by “propagate rewards back to early states”? Could you elaborate on this?
> >
> > Also, I agree with Reviewer Nqr6 that results on hard exploration tasks may provide further insights into the effect of incorporating DEXR into existing intrinsic motivation methods.

---

### Official Review · Reviewer_Nqr6 · 2023-11-02

**Soundness:** 2 fair
**Presentation:** 2 fair
**Contribution:** 2 fair
**Rating:** 3
**Confidence:** 3

**Summary:**

This paper aims to address the problem that existing approaches to exploration with intrinsic bonuses are highly sensitive to hyperparameters that are used to balance exploration and exploitation.

This paper proposes a framework called DEXR, which can be applied to explore using intrinsic rewards in off-policy settings. Concretely, separate exploration and exploitation policies are maintained, and actions are selected through the exploitative policy till a stochastic truncation point, after which the exploratory policy takes over.

Experiments show that the proposed approach enables learning good policies in fewer samples for a wide range of exploration factors/intrinsic reward coefficients.

**Strengths:**

**S1.** Using intrinsic rewards is a popular choice for exploring environments with sparse rewards and high-dimensional state spaces. Thus, improving the hyperparameter sensitivity of these approaches and enabling applicability even in dense reward settings would interest RL practitioners.

**S2.** The proposed approach is conceptually simple and easy to integrate with various bonuses and RL algorithms.

**Weaknesses:**

**W1.** While the proposed approach shows robustness to choices of the intrinsic reward coefficient ($\beta$) in the considered environments, it is currently unclear if this approach would hinder existing curiosity-based approaches in the settings where they have shown significant benefits.

The environments are not the best representation of when curiosity-based approaches succeed. The environments considered are low-dimensional, without a hard-exploration challenge.

While I agree that the maze navigation environment is technically a sparse-reward setting, the considered mazes are ‘simple’ to explore. Even TD-3, without any intrinsic bonus, reliably achieves rewards in three of the four settings and reaches the goal sometimes in the largest maze. Many intrinsic bonus approaches have enabled success in settings where naive exploration (as in standard TD-3)  is far slower and (almost) never succeeds in the considered interactions (e.g., mazes in MiniGrid [1] or hard-exploration atari environments [2]).

**W2.** To properly evaluate the sensitivity of approaches to the intrinsic reward coefficient, further details are needed for how the range for  $\beta$ was selected. For the navigation tasks, only two values of $\beta_s=1$ and $\beta_l=10000$ were selected, and they are extremely wide apart. I have similar concerns for the Mujoco experiments, which use five values of $\beta$ in the same range. Is there a reason why lower values of $\beta$ were not considered?

**W3.** It would also be useful to understand DEXR’s sensitivity to truncation probability p. No results are presented regarding this.
Further, is there a reason why p is chosen as 1- gamma?

It would also be better to mention that p is decayed in the main text (not just in the appendix).


**W4.** The paper misses discussions regarding Bayesian RL approaches (see [3] and references within), which can elegantly balance exploration and exploitation without ugly coefficients to balance intrinsic and extrinsic rewards. There also exist some scalable variants (e.g., [4]).

On a separate note regarding presentation, the paper could be improved by incorporating a separate section for the theoretical results. I also felt that the introduction and the related work section (that immediately follows the introduction) could jointly be compressed as there is quite a bit of overlap between them.

—------------------—------------------—------------------—------------------—------------------

### References

[1] Chevalier-Boisvert, M., Dai, B., Towers, M., de Lazcano, R., Willems, L., Lahlou, S., ... & Terry, J. (2023). Minigrid & Miniworld: Modular & Customizable Reinforcement Learning Environments for Goal-Oriented Tasks. arXiv preprint arXiv:2306.13831.

[2] Bellemare, M., Srinivasan, S., Ostrovski, G., Schaul, T., Saxton, D., & Munos, R. (2016). Unifying count-based exploration and intrinsic motivation. Advances in neural information processing systems, 29.


[3] Ghavamzadeh, M., Mannor, S., Pineau, J., & Tamar, A. (2015). Bayesian reinforcement learning: A survey. Foundations and Trends® in Machine Learning, 8(5-6), 359-483.

[4] Osband, I., Blundell, C., Pritzel, A., & Van Roy, B. (2016). Deep exploration via bootstrapped DQN. Advances in neural information processing systems, 29.

**Questions:**

Q1. Regarding Figure 2, could the authors explain why DEXR’s exploration is preferable to standard intrinsic motivation? It seems like DEXR would not help in reward-free exploration.

Q2. Since off-policy approaches can be more sample-efficient, I would like to understand if it is possible to use TD-3 + EIPO (instead of PPO) with a similar motivation, i.e., alternating interactions with exploratory and exploitative policies. Or are there other reasons why PPO needs to be used with EIPO?

Q3. Regarding the results presented in Figure 6, are similar figures available for the case where performance is not aggregated across choices of intrinsic reward coefficients ($\beta$)?

---

> ### Author Response · Authors · 2023-11-20
>
> We would like to thank the reviewer Nqr6 for the suggestions and the questions. We address the concerns and the questions in the following:
>
> Q1: "Regarding Figure 2 ... not help in reward-free exploration."
>
> A1: Thanks for raising this point. We would like to clarify that DEXR is primarily designed to enhance stability across hyperparameters,  and thus enable efficient exploration in an environment-agnostic manner, rather than to maximize exploration in environments without reward. Nevertheless, despite it is not designed for the reward-free setting, our theoretical analysis shows that it would perform on par with standard intrinsic reward exploration algorithms, which indicates that DEXR can also explore efficiently under the reward-free setting per the findings in [1].
>
> Q2: "Since off-policy approaches can be more sample-efficient ... PPO need to be used with EIPO?"
>
> A2: Thank you for raising the thoughtful question about using TD3 as an alternative backbone for EIPO. You make an excellent point that off-policy algorithms could provide advantages. There were two key factors in preserving the PPO backbone in EIPO:
> 1. EIPO relies on accurate policy value estimates to switch between exploration/exploitation. As an on-policy method, PPO better fits this need.
> 2. Significantly changing the original PPO-based EIPO risks invalidating the baseline.
>
> Q3: "Regarding the results presented in Figure 6 ... reward coefficients $\beta$?"
>
> A3: We apologize for missing the split plots, we have already restored the split plots in Appendix A to provide more detailed information on the experimental results.
>
> Concern 1: "While the proposed approach shows robustness ... curiosity-based approaches succeed."
>
> Response 1: We would like to acknowledge that harder exploration problems exist, such as Montezuma’s Revenge. However, we believe our experiment, along with our theoretical analysis, supports our technical claim about DEXR well, as our focus is to enhance to stability of curiosity-driven exploration methods under different reward sparsity settings.
>
> Concern 2: "The environments considered ... the goal sometimes in the largest maze."
>
> Response 2: The reason why we chose to work with MuJoCo is two-fold:
> 1. The aim of DEXR is not to outperform the state-of-the-art in the most popular benchmark but to enhance the stability of existing curiosity-driven exploration algorithms.
> 2. As Suggested in [3], vision representation can affect performance heavily in pixel-based environments. We evaluate DEXR on MuJoCo to eliminate other factors.
>
> PointMaze is a good testbed for exploration:
> 1. Its observation space contains clear location information, which allows for monitoring visualizing, and analyzing the behavior of the agents.
> 2. The hardness of an environment should not be defined by whether the random exploration can get the reward, but defined as “how many samples are needed to find the near-optimal policy”. For reference, without structured exploration, PPO can obtain a considerable amount of reward in Montezuma’s Revenge.
>
> Despite TD3 reaching the goal location a few times in the training process in Large-Maze, still fails to escape the suboptimality. And TD3 gets stuck with its sub-optimal policy, it takes at least 2x longer for TD3 to reach the goal compared to DEXR in all tasks.
>
> Concern 3: "To properly evaluate ... why lower values of $\beta$ were not considered?"
>
> Response 3: For PointMaze, the intention was to investigate the impact of small v.s. large intrinsic reward coefficients on the performance and the behaviors of different algorithms.
>
> For MuJoCo locomotion, to evaluate the robustness of DEXR in MuJoCo locomotion tasks, we test various hyperparameters. The exploration & exploitation balance with a higher intrinsic reward coefficient is crucial: higher values encourage exploration in unknown environments but can lead to distractions and unstable optimization. We assess DEXR's effectiveness with coefficients ranging from the most natural choice 1.0 and increase up to 1000.0, examining its ability to handle these challenges.
>
> Concern 4: "It would also be useful ... why p is chosen as 1- gamma?"
>
> Response 4: We have also added some discussion on this design choice in Section 4 for more information, and provide a set of experiments on evaluating how DEXR would respond to different initial truncation probability $p$ and different decay schedules, which can be found in Appendix A.
>
> Concern 5: "The paper misses .. variants (e.g., [4])".
>
> Response 5: We have already added some discussion on the Bayesian RL approach in the related work section.
>
> Reference:
>
> [1]Wang et al. (2020) On Reward-Free Reinforcement Learning with Linear Function Approximation.
>
> [2] Bellemare et al. (2016) Unifying count-based exploration and intrinsic motivation.
>
> [3]Kostrikov et al. (2020) Image augmentation is all you need: Regularizing deep reinforcement learning from pixels.

---

> ### Comment · Reviewer_Nqr6 · 2023-11-21
>
> Thank you for your answers/clarifications. I appreciate the time and effort spent in crafting the reponse. I will maintain my score as some of my main concerns still remain open.
>
> - Thanks for the clarification regarding Q1, it was slightly confusing since I was trying to see how DEXR better explored in that setting.
>
> - I still think applying the idea of EIPO with TD3 could be useful, but I can accept that it is probably not essential due to the reasons you mention.
>
> - Regarding responses 1 and 2, I still feel that harder exploration environments should also have been considered to understand DEXR comprehensively. It is important to know how much of a potential downside (or not) there could be from DEXR in such environments. Harder environments have been used by the considered baselines for stably training with intrinsic rewards. EIPO evaluates in the atari suite, DERL studies DeepSea. About the large point-maze, I agree that obtaining rewards should not be a measure of hardness in general, but I think it is not unreasonable for a sparse reward navigation task. Even if we consider it a relatively hard exploration task, it is still only one example on the simpler side.
>
> - Regarding response 3, could the authors clarify why $\beta=1$ is a natural choice on the lower side? Many (if not most) approaches to intrinsic exploration often use a far smaller value of $\beta$, for eg. see appendix A2 of RIDE (https://arxiv.org/pdf/2002.12292.pdf), which uses coefficients of 0.005 in some cases. The range over which the approaches are evaluated should definitely contain the best values of the intrinsic bonus coefficient. Is that the case in these experiments? Of course, the best value could differ across environments, and that would be a strong motivation for DEXR.

---

> > ### Author Response · Authors · 2023-11-22
> >
> > We thank reviewer Nqr6 for the suggestions.
> > * We will make sure to include experiments on the suggested tasks.
> > * To clarify, 0.005 is not a "natural" choice. $\beta=1.0$ is a natural choice in the sense that it means there is no hyperparameter used, the intrinsic reward is simply added to the objective, without any scaling. We will make sure to include more experiments next time, specifically, the standard exploration method works poorly with $\beta=1.0$ using Dynamics intrinsic reward, which suggests that a lower coefficient is preferred.

---

### Official Review · Reviewer_CCrp · 2023-11-04

**Soundness:** 3 good
**Presentation:** 3 good
**Contribution:** 3 good
**Rating:** 6
**Confidence:** 4

**Summary:**

This paper introduces a framework for handling the exploration-exploitation problem in reinforcement learning. This framework involves an exploitative and exploratory policy, each of which is learnt using off-policy RL using data collected by exploiting in the first few steps of an episode, followed by exploration. The paper suggests that this helps them avoid massively out-of-distribution samples that arise in off-policy RL with random exploration, and leads to superior performance compared to other intrinsic-reward based RL approaches.

**Strengths:**

- The lower dependence on the success of this approach on the proportion of intrinsic reward (exploration factor $\beta$) demonstrates that the approach is robust to this hyperparameter, which is a key claim made in this paper.
- The results show that the approach works in both sparse and dense reward settings, which is also a key claim made in this paper.
- Figure 8 provides very good intuitive qualitative evidence of how DEXR is more agnostic to $\beta$ than the other baselines; I like how this paper goes beyond the normal learning curves to provide visualisations for their central claims.

**Weaknesses:**

- The related work section covers important papers, but I felt a key section was missing, the Go-explore family of algorithms. [1] proposed Go-Explore, which I think should be discussed in related work. [2] actually has quite a similar proposition to this paper; there are two policies learnt, one that learns only to exploit and one that learns only to explore. While that is in the meta-RL setup, I think it should be mentioned that there is existing work that has suggested similar ideas, but not from an intrinsic motivation perspective.
- It may also be worth covering temporal abstraction for exploration in RL in related work. For instance, [3] performs exploration at the level of options i.e. at a higher level of temporal abstraction.
- In the results, the paper mentions that DEXR enjoys notably smaller variances compared to DeRL and Exp but that does not seem to be the case (or at least does not seem to be obvious) in Figure 6. Could the authors provide cleaner plots (with only one DEXR, one Exp, and one DeRL variant, instead of the full set of plots)?
- Minor points:
    - Figure 1 has a grey line at the right; it seems to be a screenshot that didn't crop properly?
    - Algorithm 1: DXER -> DEXR
    - Page 6 – board -> broad


References:

[1] First return, then explore. Ecoffet et al, 2021.

[2] First-Explore, then Exploit: Meta-Learning Intelligent Exploration. Normal and Clune, 2023.

[3] Temporally-Extended $\epsilon$-Greedy Exploration. Dabney et al, 2020.

**Questions:**

- If you have the truncation probability, why do you need the horizon $H$ in the algorithm? Is it an implementation detail to ensure that the exploitation phase does not cover a majority of the episode?
- In Figure 6, the results are averaged over 5 random seeds and 5 values of $\beta$. It is not standard practice to average over hyperparameters, so I am curious to know why the authors went with this particular decision. Also, the paper mentions that the split over $\beta$ is in the appendix but I was unable to find it. Could the authors clarify where I can find these results?
- Bonus (not a request for experiments): I'm curious to know if the authors experimented with iterative exploitation-exploration within an episode.

---

> ### Author Response · Authors · 2023-11-20
>
> We would like to thank the reviewer CCrp for the suggestions and for bringing up insightful questions. We address the concerns and the questions in the following:
>
> Q1: If you have the truncation probability, why do you need the horizon in the algorithm? Is it an implementation detail to ensure that the exploitation phase does not cover a majority of the episode?
>
> A1: Note that H is an environment parameter of the finite horizon setting. The algorithm takes this parameter and splits the horizon into two parts randomly (via the truncation probability). The algorithm performs exploitation in the first part and exploration in the second part. Indeed, if the environment is infinite-horizon, we can set another truncation probability $1-\gamma$ for the exploration policy.
>
> Q2: In Figure 6, the results are averaged over 5 random seeds and 5 values of $\beta$. It is not standard practice to average over hyperparameters, so I am curious to know why the authors went with this particular decision. Also, the paper mentions that the split over is in the appendix but I was unable to find it. Could the authors clarify where I can find these results?
>
> A2: We have already restored the split plots in Appendix A to provide more detailed information on the experimental results, we apologize for missing these plots in the previous version. Regarding the averaging performance across both hyperparameters and seeds, our rationale was to use such averaged results over the hyperparameters to show the robustness of our method, as the performance is promising, and the standard deviation is much lower than Exp and DeRL. Per your feedback, we move the aggregated plots to Appendix A, and in the main text, we only keep the results with Disagreement intrinsic reward with $\beta=1.0$ and $\beta=1000.0$ for better clarity.
>
> Q3: Bonus (not a request for experiments): I'm curious to know if the authors experimented with iterative exploitation-exploration within an episode.
>
> A3: The paradigm of alternating exploitation and exploitation within every episode is precisely the approach taken by EIPO: EIPO utilizes PPO as the backbone and toggles between exploration and exploitation based on state-value estimates. However, as evidenced by our visualizations in Section 5, this approach is suboptimal. As we have discussed in Section 5 of the paper, the alternative policy switching makes the exploitation policy unable to identify successful trajectories and hence hampers overall performance.
>
> Concern 1: The related work section covers important papers, but I felt a key section was missing, the Go-explore family of algorithms.
>
> Response 1: Thanks for your suggestion to include a discussion on the Go-Explore work in the Related Work section. In response, we have added a discussion on the connection between our method and the Go-Explore method in the related work section.
>
> Concern 2: (First-Explore, then Exploit: Meta-Learning Intelligent Exploration) actually has quite a similar proposition to this paper; there are two policies learned, one that learns only to exploit and one that learns only to explore. While that is in the meta-RL setup, I think it should be mentioned that there is existing work that has suggested similar ideas, but not from an intrinsic motivation perspective.
>
> Response 2: Thank you for the insightful point. Indeed this is quite relevant to our paper, and it shares some similar approach with DEXR. We have added a discussion concerning this work to our related work section.
>
> Concern 3: In the results, the paper mentions that DEXR enjoys notably smaller variances compared to DeRL and Exp but that does not seem to be the case (or at least does not seem to be obvious) in Figure 6. Could the authors provide cleaner plots (with only one DEXR, one Exp, and one DeRL variant, instead of the full set of plots)?
>
> Response 3: We have put cleaner results in the main texts, and added the detailed split plots in Appendix A.

---

> ### Comment · Reviewer_CCrp · 2023-11-22
>
> I thank the authors for answering the questions I had and addressing some of the concerns particularly in the related work section. I have gone through the revised version and also concur with the concerns raised by other reviewers regarding the inclusion of another hard exploration task, and I am thus keeping my current score.

---

### Meta-Review · Area_Chair_47SC · 2023-12-05

**Metareview:**

This paper introduces a two-stage algorithm that starts with a pure exploitative policy and eventually it switches to an exploratory policy in each episode. The idea is that such an approach would lead to temporally-extended exploration and that it would reduce the sensitivity of exploration hyperparameters across different environments. I am recommending this paper to be rejected but this recommendation does not come lightly as there were conflicting opinions about this paper. Despite its positive aspects, it seems the paper would benefit from generating  additional experimental evidence to strengthen the claims made around (1) performance (additional baselines and more diverse and representative environments), as well as (2) robustness, given that there were questions about how hyperparameters were tuned and the impact of some other parameters such as p. Finally, the paper would also benefit from an (3) improved discussion around related approaches, specifically around option-based exploration methods,

**Justification For Why Not Higher Score:**

This paper would have received a higher score if it had a more extensive empirical analysis in terms of environments, baselines, and the method itself (robustness over hyperparameters and so on). It also overlooks a sizeable literature on option-based exploration.

**Justification For Why Not Lower Score:**

N/A

---

### Decision · Program_Chairs · 2024-01-16

Reject